# Bio-inspired nitric-oxide-driven nanomotor

Mimi Wan[1], Huan Chen[1], Qi Wang[1], Qian Niu[1], Ping Xu[1], Yueqi Yu[1], Tianyu Zhu[1], Chun Mao[1] & Jian Shen[1]

Current chemical-fuel-driven nanomotors are driven by gas (e.g. $H_2$, $O_2$, $NH_3$) which only provides motion ability, and can produce waste (e.g. $Mg(OH)_2$, Pt). Here, inspired by endogenous biochemical reactions in the human body involving conversion of amino acid L-arginine to nitric oxide (NO) by NO synthase (NOS) or reactive oxygen species (ROS), we report on a nanomotor made of hyperbranched polyamide/L-arginine (HLA). The nanomotor utilizes L-arginine as fuel for the production of NO both as driving force and to provide beneficial effects, including promoting endothelialisation and anticancer effects, along with other beneficial by-products. In addition, the HLA nanomotors are fluorescent and can be used to monitor the movement of nanomotors in vivo in the future. This work presents a zero-waste, self-destroyed and self-imaging nanomotor with potential biological application for the treatment of various diseases in different tissues including blood vessels and tumours.

[1] National and Local Joint Engineering Research Center of Biomedical Functional Materials, School of Chemistry and Materials Science, Nanjing Normal University, 210023 Nanjing, China. These authors contributed equally: Mimi Wan, Huan Chen. Correspondence and requests for materials should be addressed to C.M. (email: maochun@njnu.edu.cn) or to J.S. (email: jshen@njnu.edu.cn)

Micro/nanomotors have been regarded as one exciting research field owing to their huge potential in biomedical application[1–3]. Up to now, they are divided into three categories according to their propulsion modes: physical, biological, and chemical. Physical micro/nanomotors can be propelled by magnetic field, ultrasound, or light which need complicated actuation system to maintain their motion due to their lack of self-driven ability[4–6]. Biological micromotors based on bacterium or sperm cell may be self-driven responding to biochemical environment[7,8], yet, biologically active substances need special conditions for preservation and use. Moreover, the size of such micromotors is usually rather large (5–50 μm), failed to enter into cells to perform precise treatment. The concept of chemical self-driving force offers more application possibilities for micro/nanomotors. Chemical micro/nanomotors can move by expelling bubbles (driving force) formed by chemical reaction between the micro/nanomotors and their surroundings (fuel) such as $H_2O_2$, glucose, urea or other physiological fluids[9–11]. The most commonly used fuels and driving forces are summarized in Supplementary Table 1, which also display the reaction mechanisms and by-products of the motion process. It can be seen clearly that almost all chemical-fuel-driven (magnesium-based, platinum-based, and enzyme-based) micro/nanomotors can produce part of exhaust gas (such as $H_2$, $CO_2$, ammonia) or waste (such as $Mg(OH)_2$, Pt), which may be toxic to the human body (e.g., excess amount of $H_2$ may cause gas thrombus)[12]. Meantime, visualizing and monitoring devices during movement process is also great challenge for nanomotors design[13]. Therefore, the design of zero-waste, self-destroyed, self-imageable micro/nanomotors is of great significance. The biochemical reactions within human body involving conversion of L-arginine to NO by NOS and several kinds of ROS draw our attention. In this system, reactant L-arginine has an immunomodulatory function for preventing thymus degeneration and promoting growth of thymic lymphocytes. One product of NO has been denoted as Molecule of the Year in 1992 and its proposer won the Nobel Price for Physiology and Medicine in 1998, which is associated with several functions such as cognitive function, regulating the non-adrenergic/non-cholinergic relaxation of smooth muscle cells and acting as a therapeutic agent for tumor or promoting angiogenesis[14]. Another product of L-citrulline can improve immune system function, maintain joint function, balance normal blood sugar levels, contain rich antioxidants absorbing harmful free radicals[15,16].

Here we report a kind of NO-driven nanomotor (Fig. 1), in which medical used fluorescent hyperbranched polyamide (HPAM) and biocompatible zwitterion L-arginine were chosen to synthesize hyperbranched polyamide/L-arginine (HLA) nanomotors. ROS in vivo were simulated with $H_2O_2$, and the kinetic behavior of nanomotors was studied in vitro. Then, the motion behaviors of nanomotors in $H_2O_2$ aqueous solution and cell environment, cell uptake behaviors of nanomotors and the influence of nanomotors on the cells (Michigan cancer foundation-7 (MCF-7) and human umbilical vein endothelial cells (HUVECs)) were investigated in detail. Finally, we extend this nanomotor family by using different matrix materials (chitosan, polylysine, and heparin/folic acid with amiogroup) to react with L-arginine. Thus, a truly zero-waste and self-destroyed nanomotors are constructed, in which both the reactant and the products can have a beneficial effect on human body. The self-imaging of the nanomotors in cellular condition is realized by help of the good fluorescent property of HPAM, providing possibility of tracking the devices in vivo in the future.

## Results

**Characterizations of $HLA_n$ nanomotors.** The synthesized $HLA_n$ (where n represent the mass ratio of L-arginine to HPAM) nanomotors were first characterized by transmission electron microscopy (TEM) images (Supplementary Fig. 1), which show that the particle size of $HLA_n$ nanomotors increase with the increase of L-arginine concentration (from about 120 nm for $HLA_5$ nanomotors to several micrometers for $HLA_{20}$ nanomotors). Meantime, the $HLA_5$, $HLA_{10}$, and $HLA_{15}$ nanomotors were negatively stained with uranyl acetate and then observed using TEM (Fig. 2a–c). The resulting images reveal spherical morphologies of these nanomotors and the particle sizes for $HLA_5$, $HLA_{10}$, and $HLA_{15}$ are about 120, 170, and 385 nm, respectively, which matches the hydrodynamic diameter variation trend measured by dynamic light scattering (DLS, Supplementary Fig. 2). The statistical analyses of multiple particles were conducted by several TEM images. As shown in Supplementary Fig. 3, the obtained particle sizes after statistical analyses for $HLA_5$, $HLA_{10}$, and $HLA_{15}$ match the particle sizes shown in Supplementary Fig. 1 and also match the variation trend of hydrodynamic diameter measured by DLS (Supplementary Fig. 2).

We then used mass spectrometry (MS) to analyze the molecular identity of the different samples such as L-arginine, HPAM, L-citrulline, $HLA_{10}$ nanomotors, and HLC (HPAM/L-citrulline composite after $HLA_{10}$ nanomotors reacting with 10% $H_2O_2$ for 24 h). Results display that mass spectrum of $HLA_{10}$ nanomotors contain the main characteristic peaks of L-arginine (139.10, 157.11) and HPAM (315.25). Among them, the peak at 157.11 can be attributed to L-arginine losing one –OH, while the 115.09 peak can be regarded as L-arginine losing a group of $NH_2$–C(=NH)–NH–. In order to characterize the product after $HLA_{10}$ nanomotors reacting with $H_2O_2$, mass spectrum of HLC, the supernatant HPAM/L-citrulline composite after $HLA_{10}$ nanomotors reacting with 10% $H_2O_2$ for 24 h, was also shown in Fig. 2d–h. The peaks at 139.10 and 157.11 disappear and peak at 115.09 (ascribed to L-citrulline losing one –$NH_2$ and one –COOH functional group) appear, illustrating the successful production of L-citrulline. Owing to the strict demands for biomedical applications, self-destroyed nanomotors have been one of the ideal choice for that they will disappear after completing their tasks without causing extra damage to human body[13]. Hence, the self-destroyed process of $HLA_{10}$ nanomotors in $H_2O_2$ solution was also detected by TEM technique (Supplementary Fig. 4), which illustrate that the particle size decreases from about 170 to 70 nm for 18 h owing to the weakened interactions between HPAM and L-arginine[17].

We further characterize $HLA_n$ nanomotors using FTIR and XPS spectra (Supplementary Figs. 5–10). FTIR spectra of $HLA_{10}$ nanomotors show both the characteristic peaks of HPAM and L-arginine (a carbonyl peak at 1660 $cm^{-1}$, and 1500–1600 $cm^{-1}$ assigned as C=N bond (Supplementary Fig. 5))[18]. The comparison of the FTIR spectra of $HLA_{10}$ nanomotors before and after reacting with $H_2O_2$ show that the peak intensity at 1660 and 1380 $cm^{-1}$ (assigned to the carbonyl and amide bonds, respectively) is significantly enhanced, indicating that the C=N of L-arginine in the $HLA_{10}$ nanomotors becomes C=O bond, suggesting the reaction is successful (Supplementary Fig. 6)[19].

XPS (C1s) spectra of HPAM, L-arginine, and $HLA_{10}$ nanomotors display that the main peak positions of each sample are concentrated in three locations corresponding to C=O, C–N, and C–C bond (Supplementary Fig. 7)[20]. Comparing C1s spectrum of HPAM with that of L-arginine (Supplementary Fig. 8), the lower proportion of C=O may be due to the fact that XPS detected the distribution of functional groups on the surface of the material[21]. The terminal group of HPAM is mostly amino group, so its

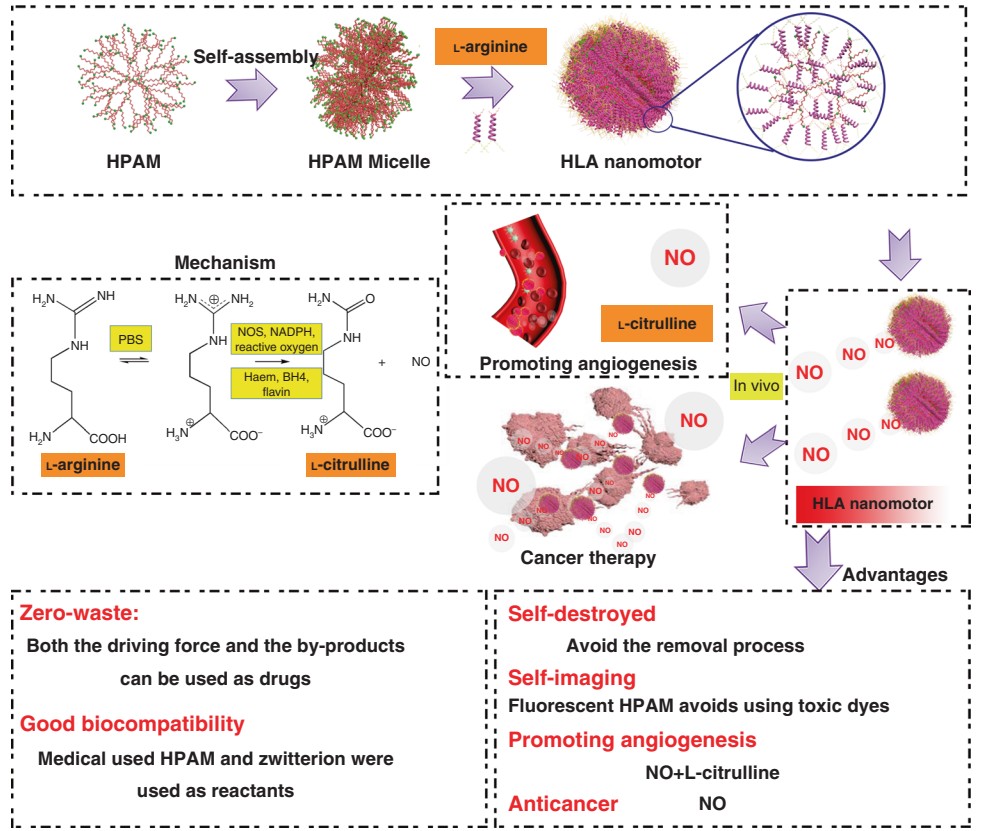

**Fig. 1** Schematic illustration of the formation of zwitterion-based nanomotor and the NO generation principle

proportion of C–N and C–C is higher than that in L-arginine. And L-arginine is rich in carboxyl groups, hence its proportion of C=O peaks is higher. Compared $HLA_{10}$ nanomotors with L-arginine, the proportion of C=O is significantly reduced, while the proportion of C–N is significantly increased, indicating that the functional groups on the surface of $HLA_{10}$ nanomotors are mostly C–N rather than C=O, which confirms the proposed mechanism of the formation of $HLA_n$ nanomotors (Fig. 1): The carboxyl group (COO⁻) in L-arginine and the amino group in HPAM are combined by electrostatic interaction to form nanoparticles, so the carboxyl group is embedded in the cavity of the HPAM structure, and the amino group at the other end is exposed outside. *XPS* (N1*s*) spectra of HPAM, L-arginine, and $HLA_{10}$ nanomotors reveal that both C–N peak from HPAM and C=N peak from L-arginine are contained in $HLA_{10}$ nanomotors, illustrating successful formation of $HLA_{10}$ nanomotors (Supplementary Figs. 9 and 10)[22].

The possible binding mechanism between HPAM and L-arginine proposed in our case is that the –COOH group in L-arginine being attracted by positive –NH₂ group, leaving –C=NH and –NH₂ groups of L-arginine locating outside of the $HLA_n$ nanomotors and retaining high reactive functional group (–C=NH) of L-arginine. In order to verify the exposed functional group of $HLA_n$ samples, FITC (Fluorescein isothiocyanate isomer I, (Supplementary Fig. 11), which can react with –NH₂ group to form covalent bond)was used to modify the surface of $HLA_{10}$ to form FITC-$HLA_{10}$. The fluorescence spectra of FITC, $HLA_{10}$, and FITC-$HLA_{10}$ were detected to characterize whether FITC can be modified on the surface of $HLA_{10}$. As shown in Supplementary Fig. 12, FITC-$HLA_{10}$ displays similar fluorescence spectrum (peak located at 510 nm) as FITC with slightly decreased peak intensity, while $HLA_{10}$ displays no peak at the wavelength of about 510 nm. These results verify the fact that the exposed functional groups of

$HLA_n$ nanomotors are –NH₂ groups, proving the suggested formation mechanism. Meantime, the highly negative charged molecule heparin (structure of heparin was shown in Supplementary Fig. 13) was also used to react with L-arginine to further prove the proposed formation mechanism. According to the proposed mechanism, the negative charged heparin can form nanoparticles with L-arginine (denoted as Hep/L-arginine) through electrostatic attraction between –COO⁻ from heparin and the –NH₂ groups from L-arginine. Hence, –NH₂ groups in L-arginine are covered by –COO⁻ from heparin. As a result, FITC cannot react with Hep/L-arginine. As shown in Supplementary Fig. 12, neither Hep/L-arginine nor FITC-Hep/L-arginine show peak at 510 nm, indicating that FITC cannot react with Hep/L-arginine, further verifying the proposed formation mechanism of $HLA_n$ nanomotors.

The fluorescence spectra of HPAM water solution (4 mg mL⁻¹) with different excitation wavelengths, HPAM water solution under different concentrations, and $HLA_n$ nanomotors were detected, respectively. From Supplementary Fig. 14a, the wavelength of 420 nm was chosen as the excitation wavelength for subsequent experiments. It can be seen from Supplementary Fig. 14b that the fluorescence intensities increase with the increasing concentrations of HPAM, and the fluorescence intensities change between 80 and 700 (a.u.), indicating good fluorescent property of HPAM for biomedical application. Meantime, the fluorescence intensities of $HLA_n$ nanomotors are about 60 (Supplementary Fig. 14c), which may provide cell imaging ability for the nanomotors. The good biocompatibility of the $HLA_{10}$ nanomotor was also confirmed by hemolysis results (Supplementary Fig. 15).

Besides, the storage operational stability and the reproducibility in the fabrication protocol of these $HLA_n$ nanomotors were also investigated. The synthesized $HLA_{10}$ nanomotors can be kept for at least 1 month (at room temperature) without sedimentation

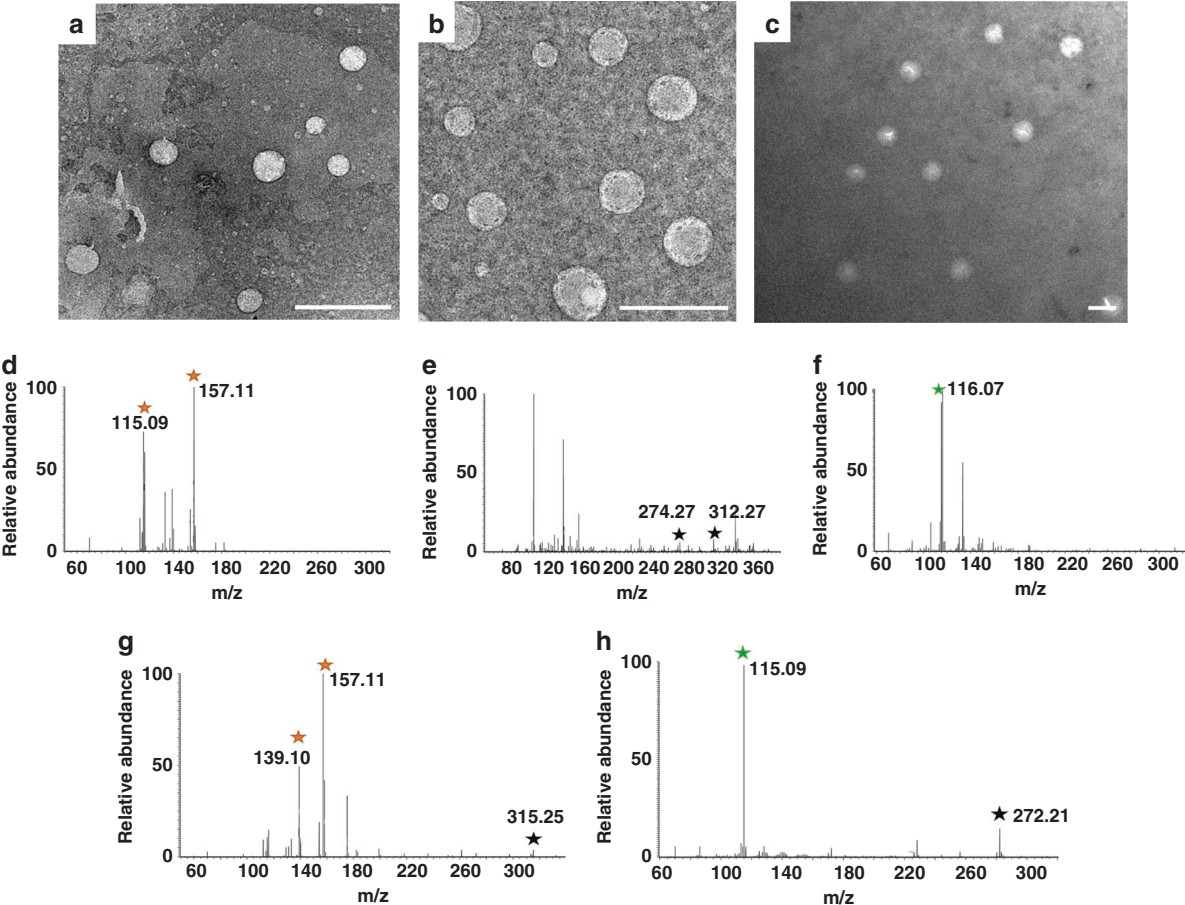

**Fig. 2** Characterizations of the nanomotors. TEM images of **a** HLA$_5$, **b** HLA$_{10}$, and **c** HLA$_{15}$ negatively stained with uranyl acetate (scale bar: 500 nm); MS spectra of **d** L-arginine, **e** HPAM, **f** L-citrulline, **g** HLA$_{10}$ nanomotor, and **h** HLC (supernatant HPAM/L-citrulline composite after HLA$_{10}$ nanomotor reacting with 10% H$_2$O$_2$ for 24 h)

(Supplementary Fig. 16) and DLS results display that HLA$_{10}$ nanomotors maintain similar particle size (Supplementary Fig. 17), indicating good storage stability of the fabrication protocol we used. Moreover, the synthesis method was repeated for five times, and DLS of HLA$_{10}$ was detected to verify good operational stability and reproducibility of the fabrication protocol we used. As shown in Supplementary Fig. 18, DLS results for the HLA$_{10}$ nanomotors prepared by five repeated times of experiment are similar with each other, implying good operational stability and reproducibility of this fabrication protocol we proposed.

**Motion behavior of the HLA$_n$ nanomotors.** In vivo, the presence of metabolic enzyme NOS, and its co-factors like nicotinamide adenine dinucleotide phosphate (NADPH), as well as ROS, can convert L-arginine to NO and L-citrulline. Thus in this work, H$_2$O$_2$ was used to simulate ROS to investigate the nanomotors movement behavior. The Movies with multiple particles for HLA$_n$ nanomotors were taken, and the results were presented and summarized in Supplementary Movie 1 and Supplementary Fig. 19. It can be seen from the results that when plenty of particles appear in one Movie, large numbers of bubbles are continuously generated, which would greatly influence the study of the movement behavior of a single nanomotor. Hence, a single particle was chosen as the study object to investigate its movement behavior. Time-lapse images of HLA$_n$ nanomotors in 10 s (20% H$_2$O$_2$), taken from Supplementary Movie 2, was shown in

Fig. 3a–d. These images illustrate that HLA$_5$ nanomotors cannot move in 20% H$_2$O$_2$ solution, while HLA$_{10}$ nanomotors can move with the generation of bubbles on its one side with straight movement line. For HLA$_{15}$ nanomotors and HLA$_{20}$ nanomotors with higher concentration of L-arginine and larger size, the bubbles are generated in several directions and the movement line is a curve. Meantime, time-lapse images (Supplementary Movie 3) displaying the tracking trajectories of HLA$_{10}$ nanomotors in 10 s under different concentration of H$_2$O$_2$ were summarized in Fig. 3e–g, which display that the movement lines of HLA$_{10}$ nanomotors are straight lines in different concentrations of H$_2$O$_2$. Based on the preparation method, L-arginine is evenly distributed on the outside of the HLA$_{10}$ nanomotors, so the small bubbles can be generated in all sites of the nanomotors. These small bubbles will quickly aggregate to form a large bubble due to the small size of the nanomotors. Therefore, it can be seen from the movie that the bubble gradually aggregate and increase with reaction time. When the bubble grows on the nanomotors surface, the nanomotors will move away from the center of the bubble owing to the bubble growth force. The bubble will suddenly disappear once it reached a maximum radius[23], then the nanomotors can be driven after the pressure in large bubbles being high enough to overcome the surface energy of the gas–liquid interface. The nanomotors can continually generate small bubbles, and these small bubbles are subject to the friction of the moving nanomotors. The friction direction is opposite to the direction of movement of the nanomotors. As shown in Supplementary Fig. 20, the small bubbles accumulate in the

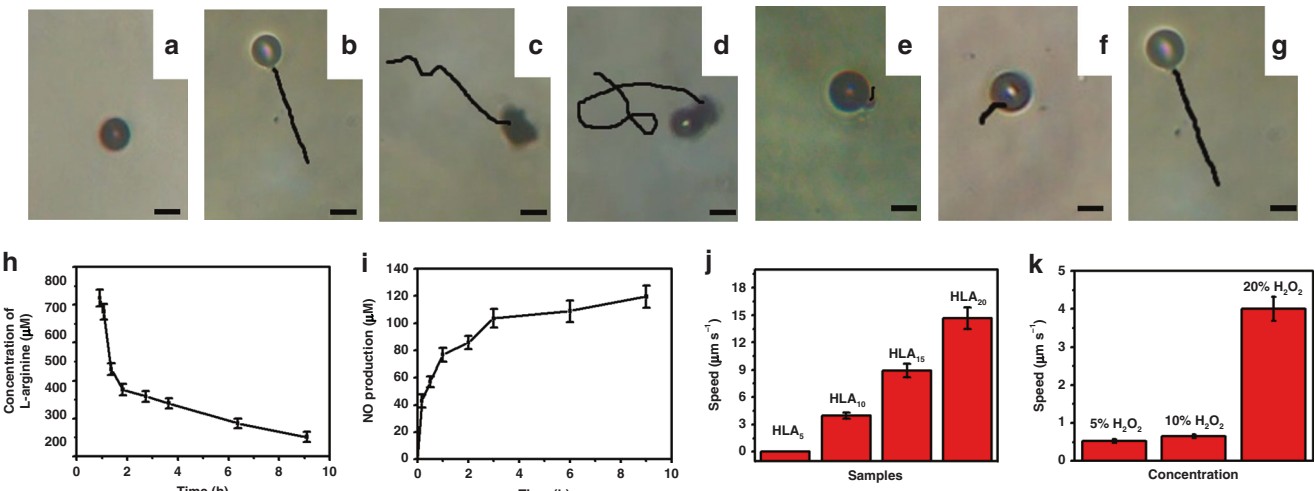

**Fig. 3** Movement behavior of the nanomotors. Time-lapse images (Supplementary Movie 2) displaying the tracking trajectories of **a** HLA$_5$, **b** HLA$_{10}$, **c** HLA$_{15}$, and **d** HLA$_{20}$ nanomotor in 10 s (20% H$_2$O$_2$, scale bar: 5 μm); Time-lapse images (Supplementary Movie 3) displaying the tracking trajectories of HLA$_{10}$ nanomotor in 10 s under **e** 5%, **f** 10%, and **g** 20% of H$_2$O$_2$ (Scale bar: 5 μm); **h** The consumption of L-arginine and **i** production of NO during motion process for HLA$_{10}$ nanomotor (20% H$_2$O$_2$); The speeds of **j** HLA$_n$ nanomotor in 20% H$_2$O$_2$ and **k** HLA$_{10}$ nanomotor in different H$_2$O$_2$ concentrations. Experimental data are mean +/−s.d. of samples in a representative experiment ($n = 3$)

friction direction to the large bubbles and continue to grow into large bubbles, thus continuing to push the nanomotors. And the subsequent bubble generation will grow in this direction. As a result, when the concentration of L-arginine is low (HLA$_5$), the generation rate (Supplementary Fig. 21, 0.023 bubble/s) of the bubbles is too small to promote its movement during the process of growing up. The rate of bubble generation (about 0.5 bubble/s for HLA$_{10}$) increases with the amount of L-arginine, which can facilitate the movement of the nanomotors. As for HLA$_{15}$ and HLA$_{20}$, the rates of bubble generation are much faster (0.6 and 0.8 bubble/s) owing to the higher amount of L-arginine in the nanomotors. Since the bubble generation speeds of HLA$_{15}$ and HLA$_{20}$ are much faster, the bubble densities on the nanomotors surface are much higher, so the small bubble will rapidly grow into a large bubble, and more than one large bubbles may be generated due to high density of small bubbles[24], so the motion direction of the nanomotor is related to the direction of the resultant force. As shown in Supplementary Fig. 20, the motion direction of the nanomotor is not a straight line but a curve (Fig. 3c, d). Based on this mechanism, we speculate that the migration of the product may be in two ways. One of the products is L-citrulline, which is produced uniformly around the nanoparticles and gradually diffuses into the surrounding solution. Another product of NO is released to form bubbles, which are uniformly generated around the nanomotors, and then gradually grow up to overcome the surface energy between gas and liquid. Therefore, the direction of migration of NO may be mainly the opposite direction of motion. The mechanisms of bubble generation and motion of nanomotors are very complicated[25], and more research will continue in the future.

L-arginine consumption and NO production amount of HLA$_{10}$ nanomotors versus reaction time in H$_2$O$_2$ solution were summarized in Fig. 3h, i. The consumption rate of L-arginine is about 17.20 μM min$^{-1}$ for the first 3 h, which decreases to 0.03 and 0.02 μM min$^{-1}$ for the following 3–6 and 6–9 h, respectively, which is the same as the trend of NO production rate, and NO production amount can reach to 120 μM at 9 h. The speeds of HLA$_n$ nanomotors increase with the concentrations of L-arginine during the synthetic process (Fig. 3j). Meantime, the speeds of HLA$_n$ nanomotors increase with the concentration of H$_2$O$_2$ (Fig. 3k). As shown in Supplementary Fig. 22, the

statistical speeds for HLA$_{10}$, HLA$_{15}$, and HLA$_{20}$ are about 3, 8, and 13 μm s$^{-1}$, respectively.

**Movement behavior of nanomotors in cellular environment.** The minimum concentration of H$_2$O$_2$ used in this case under aqueous condition is 5%. In this system, H$_2$O$_2$ is used as ROS which can be generated by cells in body. As we know, ROS, NOS, and other active components can be generated in both blood and cells[26]. Thus the concentration of H$_2$O$_2$ can be decreased in cell-experiment compared with that in aqueous condition (5%). Ninety-six-well plate was used in the cell-experiment. And the actual cell density in the body is much higher than that in 96-well plate. Thus the actual ROS in real tumor environment may be higher than that in the 96-well plate. Hence, extra H$_2$O$_2$ was introduced to simulate higher concentration of ROS in cells in actual body condition. The cancer cell used in our case is MCF-7 cell. According to the literature[27], the amount of extracellular H$_2$O$_2$ generated by one MCF-7 cell is about $2 \times 10^{-13}$ mol. In general, the number of cells in one 96-well plate for cell experiments is about $10^5$–$10^6$ with the cell culture medium volume of 0.2 mL. So we choose $5 \times 10^5$ as the average number of cells. The concentration (mass fraction, %) of H$_2$O$_2$ produced from extracellular can be calculated by the following formula:

$$C_{(H_2O_2)} = (n \times N \times M) \times 100\% / V \tag{1}$$

in which $n$ (mol) represents the amount of H$_2$O$_2$ produced by one MCF-7 cell, $N$ representing number of cells, $M$ representing the molar mass of H$_2$O$_2$, and $V$ representing the volume of the cell culture medium. The calculated concentration of H$_2$O$_2$ by MCF-7 cells in our case is about 0.002%.

Meantime, we studied the survival rate of cells at different H$_2$O$_2$ concentrations. As shown in Supplementary Fig. 23, both MCF-7 cells and HUVECs are almost unaffected when the H$_2$O$_2$ concentration is 0.002% (did not show statistically significant differences at different culture time (0.5–3 h)). So in the cell system, 0.002% H$_2$O$_2$ was used as extra ROS for cell-experiment owing to the fact that the actual cell-density in body is much greater than that in the 96-well plate under experimental conditions, which can better simulate higher concentration of ROS in the actual body condition. The choice of concentration

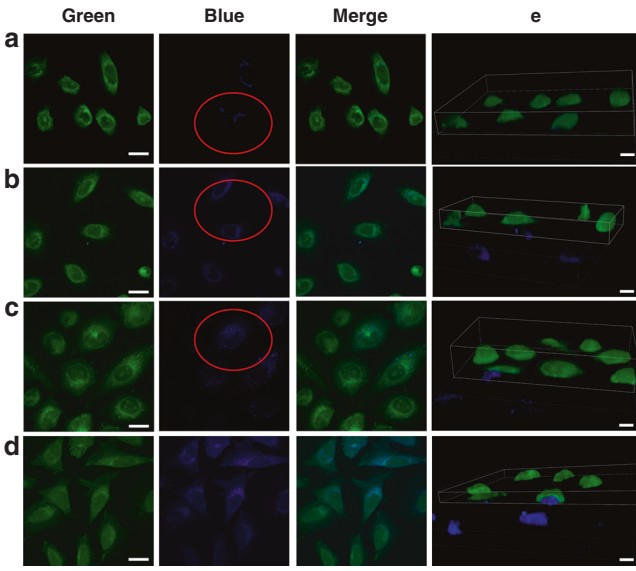

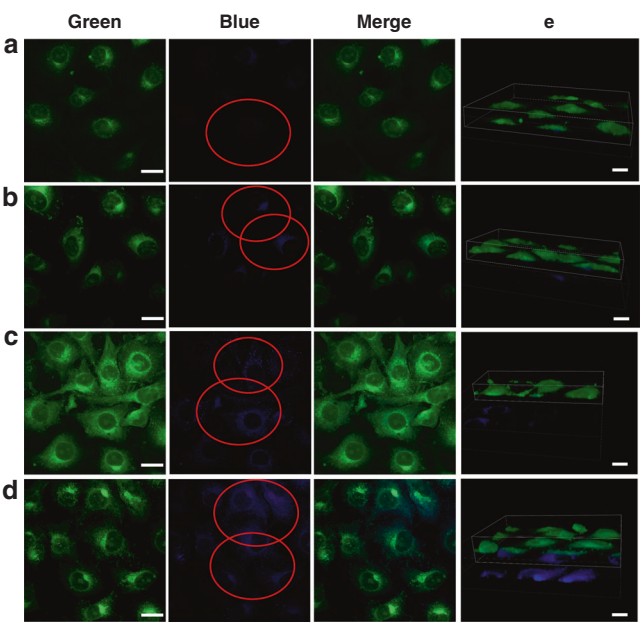

**Fig. 4** Confocal laser scanning microscopy images of the nanomotors. The cellular uptake of $HLA_{10}$ nanomotor by MCF-7 under different conditions: **a** MCF-7+$HLA_{10}$ nanomotor, **b** MCF-7+0.002% $H_2O_2$+$HLA_{10}$ nanomotor, **c** MCF-7+NOS+$HLA_{10}$ nanomotor, **d** MCF-7+NOS+0.002% $H_2O_2$+$HLA_{10}$ nanomotor, and **e** a snapshot of a 3D rendered Movie (Supplementary Movie 5) made from a stack of confocal images (Blue: nanomotors; Green: cell membrane) (Scale bar: 20 μm)

**Fig. 5** Confocal laser scanning microscopy images of the nanomotors. The cellular uptake of $HLA_{10}$ nanomotor by HUVECs under different conditions: **a** HUVECs+ $HLA_{10}$ nanomotor, **b** HUVECs+0.002% $H_2O_2$+$HLA_{10}$ nanomotor, **c** HUVECs+NOS+$HLA_{10}$ nanomotor, **d** HUVECs+NOS +0.002% $H_2O_2$+$HLA_{10}$ nanomotor, and **e** a snapshot of a 3D rendered Movie (Supplementary Movie 6) made from a stack of confocal images. (Blue: nanomotors; Green: cell membrane) (Scale bar: 20 μm)

(0.002%) is according to two reasons. One is that the selected concentration of $H_2O_2$ is similar with the $H_2O_2$ concentration produced by cells themselves. Another is that the cell viability is almost unaffected when the $H_2O_2$ concentration is 0.002% (Supplementary Fig. 23). The movement behavior of nanomotors in the cellular environment, the uptake behaviors of $HLA_{10}$ nanomotors by MCF-7 under different conditions (with or without extra $H_2O_2$ (0.002%)) were also investigated.

**Cell uptake of $HLA_{10}$ nanomotors**. The localization of nanomotors in cells is also rather important for precise treatment while most fluorescent dyes cannot be used in biomedical application due to their toxicity. Hence, it is of great need to develop nanomotors with fluorescence property. In this case, the uptake behaviors of $HLA_{10}$ nanomotors by MCF-7 and HUVECs were studied. Confocal laser scanning microscopy was used to observe the movement behavior of nanomotors in the cellular environment (Supplementary Fig. 24, Supplementary Movie 4)[25]. Surprisingly, it can be observed that the nanomotors can move in the MCF-7 cell environment with and without extra-addition of $H_2O_2$ while the nanomotors display only Brownian movement in the pure aqueous environment. Obviously, the movement of nanomotors becomes much more vigorous in the solution containing 0.002% $H_2O_2$ and MCF-7. It can also be seen that the nanomotors still maintain a certain vibration after entering the cells due to the existence of ROS, NOS, and other active components in the cells.

Besides, the uptake behaviors of $HLA_{10}$ nanomotors by MCF-7 under different conditions (MCF-7+$HLA_{10}$ nanomotor, MCF-7+0.002% $H_2O_2$+$HLA_{10}$ nanomotor, MCF-7+NOS+ $HLA_{10}$ nanomotor, MCF-7+NOS+0.002% $H_2O_2$+$HLA_{10}$ nanomotor,) were also investigated (Fig. 4, Supplementary Movie 5). As shown in Fig. 4, the morphology of MCF-7 is not affected by the addition of 0.002% $H_2O_2$. After co-culture with $HLA_{10}$ nanomotors (without extra $H_2O_2$ addition, 3 h), a small amount

of fluorescent particles appear in the MCF-7 cells. When MCF-7 is incubated with $HLA_{10}$ nanomotors with extra $H_2O_2$ (0.002%), the amount of fluorescent particles in the cells are enhanced. Moreover, the existence of NOS can also greatly promote the cell uptake of the nanomotors. And MCF-7 cells with both extra $H_2O_2$ (0.002%) and NOS addition display the largest amount of nanomotors in cells. Results indicate that the increase of concentration of ROS and NOS can promote the nanomotors movement, thereby promoting uptake of the nanomotors by the cells. The uptake behaviors of $HLA_{10}$ nanomotors by HUVECs under different conditions also display similar trend (Fig. 5, Supplementary Movie 6).

In order to determine whether the nanomotors are inside or on the surface of the cells, the confocal laser scanning microscopy and corresponding 3D reconstruction images were further used to characterize the location of these nanomotors[28]. Microscope images were taken under a 100× magnification oil objective using confocal laser scanning microscopy. In addition to capturing normal images, Z-stacks were recorded to obtain an orthogonal view of the cells and 3D images, in which green color represented cell membrane and blue color represented the nanomotors (Figs. 4e, 5e). By cutting the green channel of the 3D images, it can be observed that the green cell membrane is gradually peeled off, exposing the internal blue color ($HLA_{10}$), confirming the intracellular localization of the $HLA_{10}$ nanomotors. Results of Figs. 4e, 5e can confirm the intracellular localization of the nanomotors. The viabilities of HUVECs after HLA nanomotors intake (incubation of HUVECs with $HLA_{10}$ nanomotors under 0.002% $H_2O_2$) at different times were detected by MTT method. As shown in Supplementary Fig. 25, the cell viability of HUVECs after HLA nanomotors (under 0.002% $H_2O_2$) intake gradually increases with the cultured time (1–7 days).

Meantime, in order to further confirm the effect of $H_2O_2$ on the uptake behavior of nanomotors by cells, we measure the NO

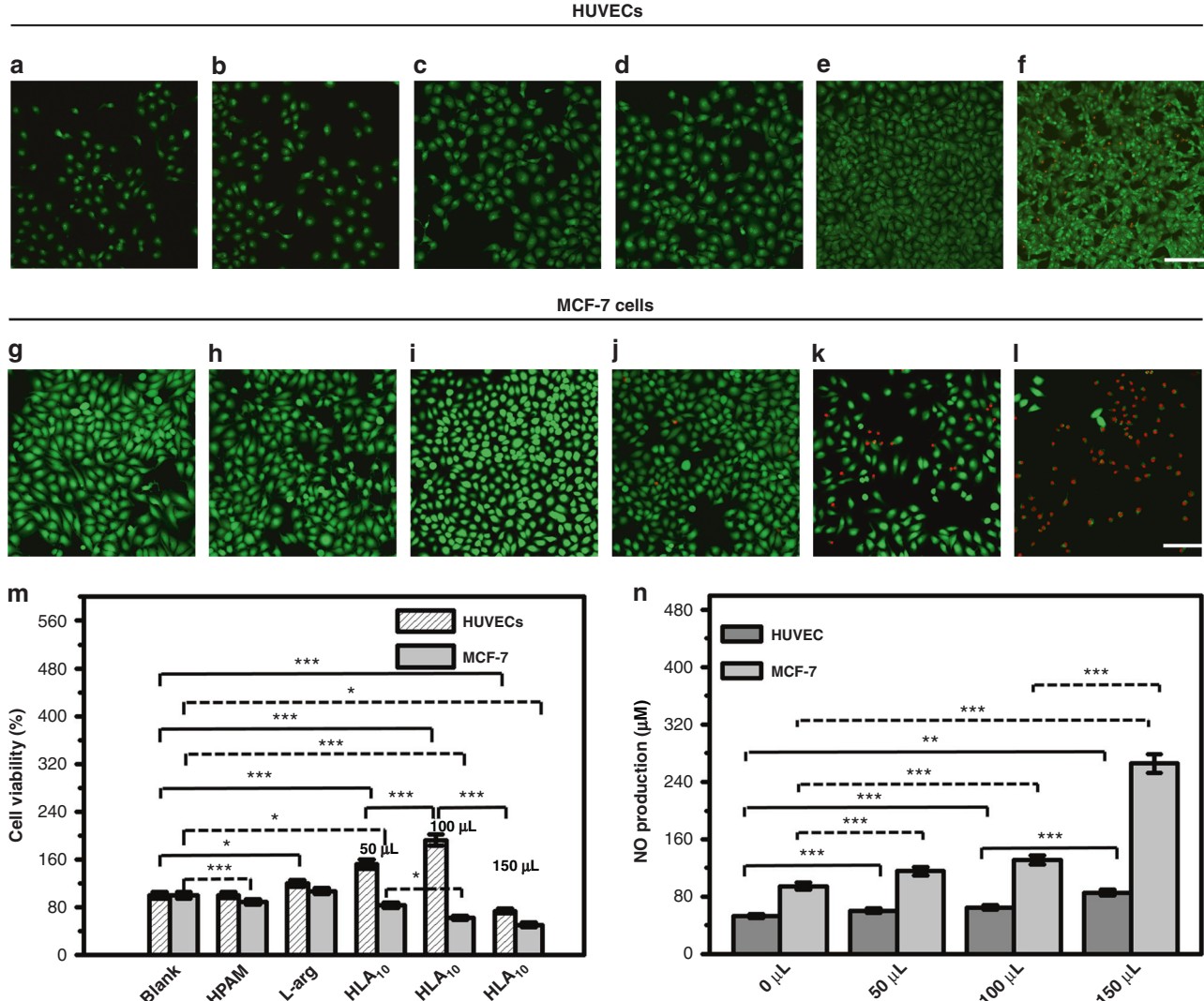

**Fig. 6** Cell viabilities tests. Live/Dead cell images of **a**, **g** control cell, **b**, **h** cell+HPAM, **c**, **i** cell+L-arginine, **d**, **j** cell+HLA$_{10}$ nanomotor 50 μL, **e**, **k** cell+HLA$_{10}$ nanomotor 100 μL, **f**, **l** cell+HLA$_{10}$ nanomotor 150 μL (Scale bar: 100 μm); **m** MTT results the samples for different cells; **n** NO production amount during the co-culture of different amounts of HLA$_{10}$ nanomotor with HUVECs and MCF-7 cells for 4 h. Asterisk (*) denotes statistical significance between bars (*$p < 0.05$, **$p < 0.01$, ***$p < 0.001$) using one-way ANOVA analysis. Experimental data are mean+/−s.d. of samples in a representative experiment ($n = 3$)

ratio of intracellular and extracellular environment after the nanomotors been uptaken by MCF-7 cells under different conditions (with or without H$_2$O$_2$) (Supplementary Fig. 26). About 73% of the produced NO is generated in MCF-7 cells, and about 79% of NO is produced in cells with higher ROS (0.002% H$_2$O$_2$). This phenomenon may be due to the fact that higher level of ROS can result in an increase in the rate of movement of nanomotors, then the uptake of nanomotors by MCF-7 is enhanced.

**Influence of HLA$_{10}$ nanomotors on the cells (HUVECs and MCF-7).** To investigate the effect of HLA$_n$ nanomotors on cell viability, the effects of different concentrations of HLA$_{10}$ nanomotors on the growth of HUVECs and MCF-7 were studied in detail (Supplementary Figs. 27 and 28). As shown in Fig. 6a–l, HPAM alone (100 μL, 12 h) has no effect on the growth of HUVECs and MCF-7 and in fact, L-arginine alone promotes the cells proliferation. The addition of 50 or 100 μL HLA$_{10}$ nanomotors also promote the cells growth with a significant increased cell density without cell-death. However, a higher amount of

(150 μL) HLA$_{10}$ nanomotors causes many dead cells, indicating that excess HLA$_{10}$ nanomotors can inhibit cell growth. The results of MTT also give the similar trend (Fig. 6m), which illustrate that 100 μL of HLA$_{10}$ nanomotors can increase the cell viability by nearly two times (no statistically significant was showed between HPAM and blank cells by one-way ANOVA). The NO production amount was detected and summarized in Fig. 6n, certain amount of NO can be produced in the cultured HUVECs for even without the addition of H$_2$O$_2$, which increase with the increase amount of HLA$_{10}$ nanomotors. In particular, when the HLA$_{10}$ nanomotors addition amount is 150 μL, NO release amount increase to 240 μM (12 h, Supplementary Fig. 29), such a high level of NO may cause cytotoxic effects[29]. The fluorescence images of living and dead cells of MCF-7 display dead cells with HLA$_{10}$ amount of 100 μL (Fig. 6k), in which condition the NO production amount is about 130 μM, much higher than that in HUVECs (65 μM). The possible reason is that the ROS in the cancer cell is higher than normal cell, so the produced NO amount in cancer cell is higher than that in normal cell[30]. Hence, HLA$_{10}$ nanomotors can be used as anticancer drug

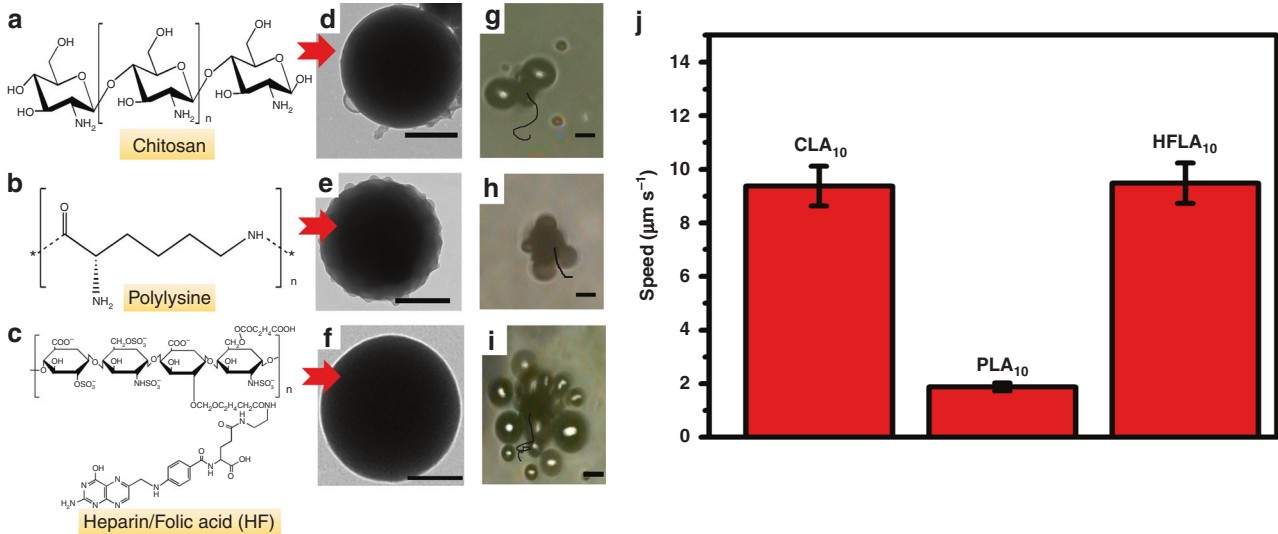

**Fig. 7** Characterizations of nanomotors with other components. Nanomotor with their counterpart chemical structure of **a** chitosan, **b** polylysine, **c** heparin/folic acid; TEM images of **d** CLA$_{10}$ nanomotor, **e** PLA$_{10}$ nanomotor, and **f** HFLA$_{10}$ nanomotor (Scale bar: 200 nm); Time-lapse images (Supplementary Movie 7) displaying the tracking trajectories in 10 s (10% H$_2$O$_2$) of **g** CLA$_{10}$ nanomotor, **h** PLA$_{10}$ nanomotor, and **i** HFLA$_{10}$ nanomotor (Scale bar: 5 μm); **j** The movement speed of the different nanomotors. Experimental data are mean +/−s.d. of samples in a representative experiment (n = 3)

for its ability to produce NO continuously. Especially, the amount of NO produced in MCF-7 (265 μM) is significantly higher than that in HUVECs (85 μM) with the 150 μL of the added HLA$_{10}$ (4 h). The amount of NO produced in cancer cells is higher because of a more efficient movement of the HLA$_{10}$ nanomotors due to the higher concentration of ROS in these MCF-7 cells than that in HUVECs, in which one MCF-7 cell can produce $2 \times 10^{-13}$ mol of H$_2$O$_2$[27]. Digital Ca$^{2+}$ imaging of the cultured HUVECs with the addition of HLA$_{10}$ nanomotors was detected (Supplementary Fig. 30). The fluorescence ratio of the nanomotors increases a lot after 200 s, which is not observed on HPAM and L-arginine. After another period of 300 s, all cells return to a similar plateau level of fluorescence ratio by washing the cells with Ca$^{2+}$ buffer, indicating that NO produced by nanomotors after they enter the cell can cause the calcium influx of cells, and this change is reversible.

In order to verify the universality of the formation mechanism of HLA$_n$ nanomotors proposed in this case, that is, the positively charged amino-enriched organic substances and the negative carboxyl groups in L-arginine combined through weak electrostatic force to form nanoparticles, three amino-enriched organic compounds (chitosan, (Mw. 3000–6000 Da), polylysine (Mw. 3000–4000), heparin/folic acid (FA) (Mw. 5000–10,000)) were chosen to react with L-arginine, and the morphology/movement behavior of the obtained nanoparticles (named as CLA$_{10}$ nanomotors, PLA$_{10}$ nanomotors, HFLA$_{10}$ nanomotors) were investigated (Fig. 7 and Supplementary Movie 7). As shown in Fig. 7a–f, all particle sizes of CLA$_{10}$ nanomotors, PLA$_{10}$ nanomotors, and HFLA$_{10}$ nanomotors are larger than 500 nm, and the speeds of CLA$_{10}$ nanomotors and HFLA$_{10}$ nanomotors with high molecular weight reactant are much higher than that of HLA$_{10}$ nanomotors with HPAM (Mw. 1000) as reactant (Fig. 7g–j). Results confirm the universality of this mechanism we proposed, which can be extended to synthesize various kinds of nanomotors expelled by NO.

## Discussion
As we know, folic acid receptors (FRs) display limited expression on healthy cells but often present in large numbers on cancer cells

surface[31]. Therefore, FA is usually regarded as an important molecule for FR-mediated targeted delivery for anticancer drugs. In our case, heparin was used to react with FA to form HF (Heparin/FA) NPs[32], then HF NPs were used to form HFLA$_{10}$ NPs (heparin/folic acid/L-arginine) with L-arginine. In order to confirm the targeting effect of HFLA$_{10}$ NPs for MCF-7 cells, MTT and cell uptake of HLA$_{10}$ and HFLA$_{10}$ nanomotors co-cultured with MCF-7 cells were conducted. As shown in Supplementary Fig. 31, the cell viability of MCF-7 cells decreases to about 30% after co-cultured with HFLA$_{10}$ for 3 h, which is about 41% for HLA$_{10}$. Furthermore, the cell uptake tests of HLA$_{10}$ and HFLA$_{10}$ nanomotors after co-cultured with MCF-7 cells for 3 h were also detected by confocal laser scanning microscopy. As shown in Supplementary Fig. 32, HLA$_{10}$ and HFLA$_{10}$ nanomotors display similar fluorescence property, hence the fluorescence intensity detected by confocal laser scanning microscopy can represent the cell uptake amount of nanomotors. As shown in Supplementary Fig. 33 and Supplementary Movie 8, the fluorescence intensity of the MCF-7 cell co-cultured with HLA$_{10}$ is much lower than that of the MCF-7 cell co-cultured with HFLA$_{10}$ nanomotors, indicating good targeting effect of HFLA$_{10}$ nanomotors for MCF-7 cells.

HPAM dendrimer molecules have been demonstrated considerable efficacy in gene therapy and drug delivery[33,34]. In general, the synthetic monomers of HPAM includes many types, and the typical two groups of monomers are as follows: N,N′-cystaminebisacrylamide and 1-(2-aminoethyl) piperazine, methacrylate and ethylenediamine. For the HPAM containing disulfide linkages prepared via Michael addition polymerization of N,N′-cystaminebisacrylamide and 1-(2-aminoethyl) piperazine[35], its good biodegradable property can be obtained owing to the fact that abundant S–S bonds in their backbones can be easily cleaved in the presence of biological or chemical stimuli[36]. For the HPAM synthesized by using methacrylate and ethylenediamine as monomers, also show very good biocompatibility[37], and degradation is rather difficult to achieve owing to its stable property. Transfection of cultured cells has been reported using complexes between DNA and spherical cationic HPAM of this kind that consist of primary amines on the surface and tertiary amines in the interior. Hence, HPAM is difficult to degrade in human environment, requiring use of organic solvent and heating

condition. Even if HPAM synthesized by using methacrylate and ethylenediamine as monomers is degraded, the degradation process involves the cleavage of amide linkage instead of going back to its synthetic monomer, causing no biological toxicity[38]. Moreover, the fact that HPAM is difficult to degrade in human environment can also be proved by the difficulty of degradation of polyamide. Researchers had verified that polyamide can degrade in chlorinated water or by heating condition[39]. The HPAM used in this case is synthesized by using methacrylate and ethylenediamine as monomers. It has very good biocompatibility and stability because it is hardly degrade in the human environment. Besides, HPAM can be removed naturally by human body through glomerular filtration, since its molecular weight is lower than 15,000[40].

In summary, inspired by endogenous biochemical reaction in human body, one example of NO-expelled HLA nanomotors was proposed. In contrast to most nanomotors based on $H_2$ or $CO_2$, causing wastes during motion process, the mechanical power of nanomotors in this case is NO that can be used as drug to pick up some important tasks such as promoting revascularization and anti-tumor. The unique fluorescent properties of HLA nanomotors derived from HPAM enable us to monitor their enterance into the cells without the use of toxic dyes for staining, which is expected to trace nanomotors in vivo in the future. Besides, in order to promoting the formation mechanism of the nanomotors we proposed, different materials with similar properties were used as synthetic raw materials of nanomotors, and all of them have good kinematic properties. Such a kind of nanomotors hold great promise for the treatment of various diseases about blood vessel and tumor.

## Methods

**Materials**. Hyperbranched polyaminde (HPAM, HyPer N102) was purchased from Wuhan Hyperbranched Polymer Resins Science & Technology Co., Ltd. L-arginine, FA, and nitric oxide synthase (NOS) inducible from mouse were purchased from Sigma-Aldrich Co., Ltd. (USA). ε-Polylysine (Mw. 3500–5000) was bought from GL Biochem. Ltd.(China). Heparin sodium was obtained from Aladdin Chemistry Co., Ltd. Chitosan, Sodium hydroxide (NaOH), 2, 3-butanedione, naphthol, n-propyl alcohol and hydrogen peroxide ($H_2O_2$, 30%) were purchased from Sinopharm Chemical Reagent Co., Ltd. and used as received. NO Kit (Nitrate reductase method) was purchased from Nanjing Jiancheng Bioengineering Institute. 3-(4, 5-dimethyl-2-thiazolyl)-2, 5-diphenyl-2-H-tetrazolium bromide (MTT) was bought from Nanjing Heyao Biotech. Co., Ltd. (China). 3, 3′-Dioctadecyloxacarbocyanine perchlorate (DiO'; DiOC18) was purchased from Shanghai Yuanye Biotechnology Co., Ltd. (China). Uranyl acetate dehydrate purchased from Shanghai Yien Chemical Technology Co., Ltd. (China).

**Synthesis of HLA_n nanomotors**. The $HLA_n$ nanomotors were prepared via electrostatic self-assembly approach. Briefly, 0.5 mg mL$^{-1}$ HPAM solution was prepared by deionized water. Then, same volume of L-arginine with different concentrations (2.5, 5, 7.5, 10 mg mL$^{-1}$) were added into HPAM solution under sonication condition for 15 min. Next, the mixtures were centrifuged at 13,900 × g for 10 min. The product was washed three times by deionized water.

Meantime, chitosan, ε-Polylysine, heparin-folate nanoparticles[41] were used to react with L-arginine to form $CLA_{10}$ nanomotor, $PLA_{10}$ nanomotor, and $HFLA_{10}$ nanomotor according to similar procedure mentioned above.

**Detection of L-arginine concentration**. $HLA_{10}$ nanomotor prepared above was dissolved in 0.25 mL deionized water, and 2.25 mL 10% $H_2O_2$ was added to react at 37 °C for different time (0, 0.5, 1, 2, 3, 6, 9 h). Centrifugation was performed to obtain the lower layer. The residual L-arginine was detected according to the following procedure[42]: Five microliter of the mixture was added to the solution of indicator containing NaOH (1 mL, 1.0 M), 1-naphthol/propanol (1 mL, 0.6 M) and diacetyl/propanol (1 mL, 0.5 mol L$^{-1}$) (30 °C, 15 min). Then its absorbance was measured by ultraviolet spectrophotometer (Agilent 8453, Agilent Technologies Co., Ltd.) at 540 nm. Finally, the curve of concentration of L-arginine versus reaction time was plotted.

**Detection of NO production**. The produced NO during the reaction process was also detected according to NO kit (nitrate reductase method). Finally, the curve of NO concentration versus reaction time was plotted.

**Characterization**. The FTIR spectra were obtained by using a Varian Cary 5000 Fourier transform infrared spectrophotometer. MS spectra were tested by using DART-SVP (Ion Sense Inc., Saugus, MA, United States) ion source coupled to an Orbitrap Fusion Lumos mass spectrometer (Thermo Fisher Scientific, United States). XPS spectra was collected on Thermo Scientific ESCALAB 250Xi. The synthesized nanoparticles was dropped on the glass slide and dried, and $H_2O_2$ solution with certain concentration was added dropwise. Then the movement behavior of the nanoparticles was monitored by Olympus microscope fitted with a camera (Olympus Group, Japan). Confocal laser scanning microscopy (HP Apo TIRF 100X N.A. 1.49, Nikon, Ti-E-A1R, Japan) was used to capture cellular uptake and nanomotor-movement in cell environment. In order to make the morphology of $HLA_n$ nanoparticles better observed, uranyl acetate was used to make them dye. Briefly, the HLAn nanoparticle solution was deposited onto the copper net by dropping droplets. Five minutes later, the sample was deposited and a drop of 4% of uranium acetate stain was added to the copper grid. The sample was subsequently dried and observed using a JEM-2100 electron microscope.

**Cell culture**. The human umbilical vein endothelial cells (HUVECs, purchased from ATCC, cell NO. CRL-1730) and Michigan cancer foundation-7 (MCF-7, purchased from DSMZ, cell NO. 115) cells were cultured by using DMEM (Dulbecco's Modified Eagle's medium) containing fetal bovine serum (FBS, 10% (v/v)), penicillin (100 μg mL$^{-1}$) and streptomycin (100 μg mL$^{-1}$) at 37 °C with 5% $CO_2$.

**MTT assay**. MTT assay was used to characterize the in vitro cytotoxicity of the samples. Briefly, the cells (HUVECs and MCF-7) were seeded in 96-well plates at a density of $5 \times 10^4$ cells/well. The medium was replaced by complete medium containing HPAM, L-arginine, and the $HLA_{10}$ nanomotor with the volume of 50, 100, and 150 μL, which were co-cultured with cells for 24 h afterward. Then MTT reagent (50 μL) was added to each well to test cell viability with MTT concentration of 5 mg mL$^{-1}$. After co-culture of MTT with cells for another 4 h, the produced formazan precipitates were dissolved in N,N-dimethylformamide (DMF), which was transferred to a new 96-well plate and the absorbance was tested with a microplate reader (490 nm, Bio-Rad Co., Ltd., USA). The experiment was repeated for three times.

**Live/dead cell viability assay**. HUVECs and MCF-7 cells were seeded in culture dishes (35 mm). Then, the cells were co-cultured with HPAM, L-arginine, and $HLA_{10}$ nanomotor with various volumes (50, 100, and 150 μL) for 12 h. The cells were colored with Calcein-AM (2 μM, green color representing live cells) and Propidium Iodide (PI, 4.5 μM, red color representing dead cells), which were incubated at 37 °C for 0.5 h before observed with a laser confocal microscope.

**NO release detection of cells under different conditions**. The HUVECs and MCF-7 cells were seeded in 96-well plates ($5 \times 10^4$ cells/well) and cultured for 4 h until cell adherence. Then the produced NO was detected according to the method mentioned above under different condition as follows:

NO production by different amounts of $HLA_{10}$ nanomotor. The culture solution (DMEM) from wells was discarded, and fresh DMEM was added. Then different amounts of $HLA_{10}$ nanomotor (50, 100, and 150 μL) were added in both experiment and co-cultured with cells for 4 h.

NO production by $HLA_{10}$ nanomotor (100 μL) with and without $H_2O_2$. The culture solution (DMEM) from wells was discarded, and fresh DMEM was added for the experiment without $H_2O_2$. For the experiment with $H_2O_2$, DMEM containing 0.002% $H_2O_2$ was added as fresh culture solution. $HLA_{10}$ nanomotor (100 μL) was added in both experiment and co-cultured with cells for 4 h.

Intracellular and extracellular NO production. DMEM incubated cells for 4 h was co-cultured with $HLA_{10}$ nanomotor (100 μL) in blank well for another 4 h, and the net produced NO (subtracting the NO produced by the cells themselves) was denoted as extracellular NO production by $HLA_{10}$ nanomotor. Adhered cells were co-cultured with $HLA_{10}$ nanomotor (100 μL) for 4 h, the net produced NO (subtracting the NO produced by the cells themselves) was denoted as total NO production by $HLA_{10}$ nanomotor. The amount of intracellular NO production by $HLA_{10}$ nanomotor was obtained by the difference between total amount and extracellular NO production amount.

**The experiment of cell calcium imaging**. To study the effect of release of NO on cell calcium channel, the glass plates with diameter of 8 mm were seeded with cells and incubated overnight. Then, they were incubated 30 min with Fura-2 (specific fluorescent indicator of intracellular calcium ($Ca^{2+}$)). The adherent cells were put into the $Ca^{2+}$ perfusion channel. Then, HPAM (100 μL), L-arginine (100 μL), $HLA_{10}$ nanomotor (100 μL) were added. Collection of experimental signals with a calcium imaging system: Fluorescence inverted microscope (The excitation wavelengths are 340 and 380 nm, IX51, Olympus, Japan), digital camera (C11440 Hanmamatsu, Japan), and light source (Polychrome V, TILL, Germany).

**Intracellular localization**. In order to observe the localization of nanomotors in cells, the cell membrane was labeled with green fluorescence by DiO and $HLA_{10}$

nanomotor solution was observed under the excitation wavelength of 405 nm by confocal laser scanning microscopy.

Microscope images were taken with Nikon laser confocal microscope in 100× magnification oil lens. In addition to taking normal images, the orthogonal view of cells was received through Z-stack. Three-dimensional images were obtained. Green cell membranes were gradually peeled off and $HLA_{10}$ with blue fluorescence was exposed by clipping the green channel. This process was recorded by video software and made into video.

**The cell-uptake of $HLA_{10}$ nanomotor by cells**. HUVECs were seeded in culture dishes (35 mm) at density of $1 \times 10^5$. Then, the cells were cultured overnight. After abandoning culture medium, a fresh medium containing 0.002% $H_2O_2$ was used as experimental group. On the contrary, the medium without $H_2O_2$ was added as the control group. Then $HLA_{10}$ nanomotor (100 μL) was added and co-cultured with cells for 3 h. The uptake of $HLA_{10}$ nanomotor by cells was observed by confocal laser scanning microscopy.

MCF-7 or HUVECs were seeded in culture dishes (35 mm) at density of $1 \times 10^5$. Then, the cells were cultured overnight. The cellular uptake behaviors of $HLA_{10}$ nanomotor by cells under different conditions (cells+$HLA_{10}$ nanomotor, cells+0.002% $H_2O_2$+$HLA_{10}$ nanomotor, cells+NOS+$HLA_{10}$ nanomotor, cells+NOS+0.002% $H_2O_2$+$HLA_{10}$ nanomotor) were observed by confocal laser scanning microscopy.

**Statistical analysis**. The data are presented as the mean ± SD. The differences among multiple groups were evaluated by a one-way analysis of variance (ANOVA) followed by the Bonferroni post hoc test (assuming equal variances) or Tamhane's T2 post hoc test (without the assumption of equal variances). The statistical analyses were performed using SPSS software (version 19.0). Asterisk (*) denotes statistical significance between bars (*$p < 0.05$, **$p < 0.01$, ***$p < 0.001$).

**Reporting summary**. Further information on experimental design is available in the Nature Research Reporting Summary linked to this article.

## Data availability

The data that support the findings of this study are either providing in the Article and its supplementary information or are available from the authors upon reasonable request.

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

## Acknowledgements

The work was supported by Jiangsu Collaborative Innovation Center of Biomedical Functional Materials, National Natural Science Foundation of China (51641104, 21603105, 21571104), Research funding of Nanjing Normal University, Jiangsu Key Technology RD Program (BE2016010), China (BE2015603), Jiangsu agricultural science and Technology Innovation Fund Project (CX(14)2127), Technology Support Program of Science and Technology Department of Jiangsu Province (BE2015703), HuaianBio-functional Materials and Analysis Technology Innovation Platform (HAP201612). We thank Dr. Hong Ying Shen from Massachusetts General Hospital for helping us with English writing.

## Author contributions

M.M.W., C.M. and J.S. designed the experiments; H.C., Q.W., Q. N., P.X., Y.Q.Y. and T.Y.Z. synthesized the materials and collected the experimental data.

## Additional information

**Competing interests:** The authors declare no competing interests.

