## [Peer Review File · Nature Communications]

Reviewers' comments:

Reviewer #1 (Remarks to the Author):

In the manuscript NO: Driving force of Bio-inspired Zero-waste Nanomotor and Its Endothelialization and Anticancer Effect, the authors claim a method for preparation of self-assembled NO-driven nanomotor, which can be tracked via fluorescence and has the ability to promote angiogenesis. The topic is of general interest in the community, as it represents a new method to use the driving force of the system for biological applications. However, the novelty of the paper is not the required for the high standards of nature communications, as I do not feel that the paper will influence thinking in the field.

The discussion is presented in a very confusing way, and the conclusions offer are not consistent with the results presented.

I have main concerns related to the characterization of the structures as well as with the proposed mechanism for the propulsion.

First, that the bubble generation will follow a specific direction is a very interesting proposed mechanism for the actuation of the motors, but in the manuscript is not explained how the researchers came to this conclusion.

The data presented is too weak to claim "...nanomotors...may move in the body with higher reactive oxygen, NOS and other reactive species."

Especially the statistical analysis is missing from the results or is not clearly mention (only single particles are represented in the images and videos submitted for evaluation) and in the figure plots, many of the error bars are not recognizable.

Lastly, there are major sentence structure and grammatical mistakes throughout the manuscript. This causes major difficulties in reading and understanding the text and the results that the authors want to share.

Specific comments

1. Why is there only one particle shown in each video? Is the bubble formation because of the reaction, or is it caused by heterogeneous condensation on the surface?
2. The concentration of H₂O₂ used during the experiments is much higher than the concentration of reactive oxygen species found in the body (see for example Xiuzu Song, et. al, Pigment Cell Melanoma Res. 809–818, 2009). The concentrations of H₂O₂ used for the experiments are toxic and not compatible with in vivo experiments as is suggested by the text.
3. There are some subjective statements made throughout the paper, that require quantification (for example, authors write that the nanomotors have a good fluorescent property thanks to the HPAM component, but do not quantify what is good in this case, or compared to what. In another statement, the consumption of L-arginine is faster and then slower... these statements need to be quantified).
4. Also in the videos. The captions of the different particles that are shown states that these are HLA_n motors (n=5,10, and so on). I don't understand why is it possible to observe them with a light microscope. Why would polymers scatter so much, to make them visible under a light microscope? It seems more likely that the videos show some aggregates of the particles than just a single one as suggested by the captions.
5. The TEM characterization of the particles is not done properly. The images shown in figure 2 are not properly focused and a lot of information is missing to back up the claims made in the paper.
6. For the TEM analysis, there is also no statistical analysis of multiple particles, at least one image (could be in the SI) should show an overview of the sample. The HLA₂₀ image does not even show where the particle "ends". In the comparison of the particles being destroyed (Fig. 2C (a-e)), the size of the images should be consistent to make it easier for a real comparison.
7. Coming back to the proposed mechanism for the directionality of the bubble formation, how do the authors explain a migration of the reactions products, to one specific site of the particle? Why only to one specific site? In addition, this conclusion is contradictory to the images seen in the supporting video S1 and supporting information S8, where the HLA₁₅ structures show 2 bubbles forming on opposite sites.
8. The figures are key to understanding the entire article and the recognition of many symbols in the images is hard or even impossible. It would be necessary to improve the images before any publication is possible. This includes the fluorescent images.
9. From the images presented for the fluorescence studies, it is impossible to determine if the nanomotors are inside or on the surface of the cells. Being the particles made out of polymers, it is likely that they would be attracted to the cellular membrane, so how can the authors know this is not the case.
10. The results presented in the SI seem not ready for publication. Images that overlap each other, wrong label of in the graphs, different thickness of plot lines given for comparison.

11. The Supplementary information order is inappropriate and there is no reference to S9 in the main text.

12. The list of references is focused on very few contributors to the field of nanomotors. Although I do not doubt, that these groups are great contributors to the field, the authors might consider other key literature from different authors. As a suggestion, authors might consider work published by the groups of D. Wilson (ACS Nano, 2017, 11 (2), pp 1957–1963), Q. He (J. Am. Chem. Soc. DOI: 10.1021/jacs.8b06646), P. Fischer (Nano Lett., 2014, 14 (5), pp 2407–2412), B. Stedler (Chem. Mater., 2015, 27 (21), pp 7412–7418), A. Sen (Angew. Chem. Int. Ed. 2011, 50, 9374–9377) among many others.

Most importantly, there are many mistakes in the citations (name, surname order does not match between citations).

Further notes

1. I'm not an expert, but as far as my understanding goes, HPAM will degrade to produce acrylamide, a very toxic material. Since this is the material which the nanomotors are made of, is this not enough reason to think that the particles here presented are also very toxic?

2. To check that the formation of the particles happens as stated, one suggestion would be to modify the surface of the structures by the exposed (expected) group. This way the authors can prove the suggested formation mechanism.

Reviewer #2 (Remarks to the Author):

Authors developed here for the first time a new type of zero-waste and self-destroyed HLA nanomotors utilizing L-arginine as fuel, NO as driving force and L-citrulline as a beneficial by-product. Results presented demonstrated the potential of these HLA nanomotors as self-imaging nanomotors (due to their good fluorescence performance) and for endothelialization and anticancer effect (taking advantage of the use of NO both as the driving force and to promote angiogenesis and kill cancer cells). Authors perform an exhaustive characterization of the developed nanomotors by TEM, MS, Zeta potential, FTIR and XPS and their motion performance using different L-arginine and H₂O₂ concentrations. Results presented demonstrated also the uptake of HLA nanomotors by HUVEC cells, the possibility to kill cancer cells, due to the higher amount of NO produced by the HLA nanomotors as a result of the higher basal level concentration of reactive oxygen species in these cells, and the possibility to extend the protocol to synthesize other kinds of nanomotors expelled by NO using different amino-enriched organic compounds. Although the results presented are interesting, my major concern about this paper is to clarify the advantages of these NO-expelled artificial

nanomotors compared to (US)-powered nanowire motors, which have demonstrated excellent properties for internalization, killing selectively cancer cells within 5 min (Ref 3 in the paper) and even single cell real-time using attractive intracellular “OFF-ON” fluorescence switching mechanisms (ACS Nano 9 (2015) 6756–6764). Other concerns that should be addressed before recommending publication of this paper in this top Journal:

1. The viability of HUVEC cells after HLA nanomotors intake (incubation of HUVEC cells with HLA nanomotors under 0.2% H₂O₂) should be indicated at different times.
2. This phrase “Especially, the amount of NO produced in MCF-7 (265 μM) was significantly higher than that in HUVEC (85 μM) with 240 the 150 μL of the added HLA10 (4 h)” given in lines 238-240 required further discussion, such as... the amount of NO in cancer cells is higher because of a more efficient movement of the HLA due to the higher concentration of reactive oxygen species in these cells.
3. At present, the selectivity of these HLA to destroy cancer cells is given only by the higher basal level concentration of reactive oxygen species in the cell, authors should discuss with more detail this selectivity issue and ways to improve it, would it be feasible to functionalise these HLAs with adequate bioreceptors for targeted cell killing?
4. The storage and operational stability and the reproducibility in the fabrication protocol of these new HLA nanomotors should be described in more detail.
5. A minor concern: There are some misspelled words and missing spaces through the text, please revise and address.

Reviewers' comments:

Reviewer #1 (Remarks to the Author):

In the manuscript NO: Driving force of Bio-inspired Zero-waste Nanomotor and Its Endothelialization and Anticancer Effect, the authors claim a method for preparation of self-assembled NO-driven nanomotor, which can be tracked via fluorescence and has the ability to promote angiogenesis. The topic is of general interest in the community, as it represents a new method to use the driving force of the system for biological applications. However, the novelty of the paper is not the required for the high standards of nature communications, as I do not feel that the paper will influence thinking in the field.

Answer: We are pleased that the referee finds our work of general interest. We are also grateful to have this opportunity to clarify and substantiate the working mechanisms of work.

The discussion is presented in a very confusing way, and the conclusions offer are not consistent with the results presented.

I have main concerns related to the characterization of the structures as well as with the proposed mechanism for the propulsion.

First, that the bubble generation will follow a specific direction is a very interesting proposed mechanism for the actuation of the motors, but in the manuscript is not explained how the researchers came to this conclusion.

Answer: Thanks a lot for the reviewer's question.

According to the reviewer's question, we modified the bubble generation mechanism as follows:

Based on the preparation method, L-arginine was evenly distributed on the outside of the HLA₁₀ nanomotors, so the small bubbles can be generated in all sites of the nanomotor. These small bubbles would quickly aggregate to form a large bubble due to the small size of the nanomotor. Therefore, it can be seen from the movie that the bubble gradually aggregated and increased with reaction time. When the bubble grew on the nanomotor surface, the nanomotor would move away from the center of the bubble owing to the bubble growth force. The bubble would suddenly disappear once it reached a maximum radius (*Phys. Rev. Letters*, 2012, 109, 128305.), then the nanomotor can be driven after the pressure in large bubbles being high enough to overcome the surface energy of the gas-liquid interface. The nanomotors can continually generate small bubbles, and these small bubbles were subject to the friction of the moving nanomotors. The friction direction was opposite to the direction of movement of the nanomotors. As shown in Figure S20, the small bubbles accumulated in the friction direction to the large bubbles and continued to grow into large bubbles, thus continuing to push the nanomotors. And the subsequent bubble generation would grow in this direction. As a result, when the concentration of L-arginine was low (HLA₅), the generation rate (0.023 bubble/s, Figure S21) of the bubbles was too small to promote its movement during the process of growing up. The rate of bubble generation (about 0.5 bubble/s for HLA₁₀) increased with the amount of L-arginine, which can facilitate the movement of the nanomotors. As for HLA₁₅ and HLA₂₀, the rates of bubble generation were much faster (0.6 and 0.8 bubble/s) owing to the higher amount of L-arginine in the nanomotors. Since the bubble generation speeds of these two nanomotors (HLA₁₅ and HLA₂₀) were much faster, the bubble densities on the nanomotors surface were much higher, so the small bubble would rapidly grow into a large bubble, and more than one large bubbles may be generated due to high density of small bubbles (*J. Am. Chem. Soc.*, 2018, 140, 11902.), so the motion direction of the nanomotor was related to the direction of the resultant force. As shown in Figure S20, the motion direction of the nanomotor was not a straight line but a curve (Figure 3A).

Figure S20. Possible motion mechanism of (A) the HLA₁₀ and (B) HLA₁₅/HLA₂₀ nanomotors during bubble growth process.

Figure S21. Time-lapse images (Movie S2) displaying the bubble generation process of (A) HLA₅, (B) HLA₁₀, (C) HLA₁₅, (D) HLA₂₀ in 10 s, respectively (20% H₂O₂).

Figure 3. (A) Time-lapse images (Movie S2) displaying the tracking trajectories of HLA_n nanomotor in 10 s (20% H₂O₂, scale bar: 5 µm), (B) Time-lapse images (Movie S3) displaying the tracking trajectories of HLA₁₀ nanomotor in 10 s under different concentration of H₂O₂ (scale bar: 5 µm), (C) The consumption of L-arginine and (D) production of NO during motion process for HLA₁₀ nanomotor (20% H₂O₂), The speeds of (E) HLA_n nanomotor in 20% H₂O₂ and (F) HLA₁₀ nanomotor in different H₂O₂ concentrations.

The data presented is too weak to claim “...nanomotors...may move in the body with higher reactive oxygen, NOS and other reactive species.”

Answer: We thank the reviewer a lot for his/her valuable question.

According to the reviewer’s question, more experiments that involved nanomotors moving in cellular environment were carried out. The movement behaviors of nanomotors in the cellular environment (MCF-7+HLA₁₀ nanomotor, MCF-7+0.002% H₂O₂+HLA₁₀ nanomotor) were observed by confocal laser scanning microscopy, and the results were shown in Figure S24 and Movie S4 of our revised manuscript. It can be observed that the nanomotors can move both in the MCF-7 cell environment with and without extra-addition of 0.002% H₂O₂ while the nanomotors displayed only Brownian movement in the pure aqueous environment. Obviously, the movement of nanomotors became much more vigorous in the solution containing 0.002% H₂O₂ and MCF-7. It can also be seen that the nanomotors still maintained a certain vibration after entering the cells due to the existence of reactive oxygen, NOS, and other active components in the cells. Besides, the uptake behaviors of HLA₁₀ nanomotors by MCF-7 under different conditions (MCF-7+HLA₁₀ nanomotor, MCF-7+0.002% H₂O₂+HLA₁₀ nanomotor, MCF-7+NOS+HLA₁₀ nanomotor, MCF-7+NOS+0.002% H₂O₂ + HLA₁₀ nanomotor) were also investigated (Figure 4). As shown in Figure 4, The morphology of MCF-7 was not affected by the addition of 0.002% H₂O₂. After co-culture with HLA₁₀ nanomotors (without extra H₂O₂ addition, 3 h), a small amount of fluorescent particles appeared in the MCF-7 cells. When MCF-7 was incubated with HLA₁₀ nanomotors with extra H₂O₂ (0.002%), the amount of fluorescent particles in the cells were enhanced. Moreover, the existence of NOS can also greatly promote the cell uptake of the nanomotors. And MCF-7 cells with both extra H₂O₂ (0.002%) and NOS addition displayed the largest amount of nanomotors in cells. Results indicated that the increase of concentration of reactive oxygen can promote the nanomotors movement, thereby promoting uptake of the nanomotors by the cells. The uptake of HLA₁₀ nanomotors by HUVECs under different conditions also displayed similar trend (Figure 5).

These results indicated that the nanomotor we proposed may move directly under cancer cell environment of experiment, and the addition of 0.002% H₂O₂ can significantly promote the movement and the uptake of nanomotors by the cells. Therefore, we speculated that nanomotors may have the same move ability and uptake behaviour in a real high-density tumor cell environment of body like the condition of MCF-7 cells with extra-H₂O₂ (0.002%).

Figure S24. Time-lapse images (MovieS3) displaying the tracking trajectories of HLA₁₀ nanomotor under (A) PBS solution, (B) MCF-7 cell solution, and (C) MCF-7 cell + 0.002% H₂O₂ in 1 s captured by confocal laser scanning microscopy (scale bar: 5 μm).

Figure 4. Confocal laser scanning microscopy images of the cellular uptake of HLA₁₀ nanomotor by MCF-7 under different conditions: (A) MCF-7+ HLA₁₀ nanomotor, (B) MCF-7+0.002% H₂O₂+HLA₁₀ nanomotor, (C) MCF-7+NOS+HLA₁₀ nanomotor, (D) MCF-7+NOS+0.002% H₂O₂+ HLA₁₀ nanomotor, and (E) a snapshot of a 3D rendered Movie (Movie S5) made from a stack of confocal images (Blue: nanomotors; Green: cell membrane).

Figure 5. Confocal laser scanning microscopy images of the cellular uptake of HLA₁₀ nanomotor by HUVECs under different conditions: (A) HUVECs+HLA₁₀ nanomotor, (B) HUVECs+0.002% H₂O₂+HLA₁₀ nanomotor, (C) HUVECs+NOS+HLA₁₀ nanomotor, (D) HUVECs +NOS+0.002% H₂O₂+ HLA₁₀ nanomotor (E) a snapshot of a 3D rendered Movie (Movie S6) made from a stack of confocal images. Blue: nanomotors; Green: cell membrane.

Especially the statistical analysis is missing from the results or is not clearly mention (only single particles are represented in the images and Movies submitted for evaluation) and in the figure plots, many of the error bars are not recognizable.

Answer: We thank the reviewer a lot for his/her valuable question.

According to the reviewer's advice, statistical analyses for TEM images (Figure S1, S4) with multiple particles and motion speeds (Figure S22), Movies with multiple nanomotors (Figure S19, Movie S1) were summarized in our revised manuscript.

Figure S1. TEM images of (A) HLA₅, (B) HLA₁₀, and (C) HLA₁₅ negatively stained with uranyl acetate.

Figure S4. The statistics on particle size (obtained from TEM images, five distinct samples, 10 particles were taken from each image) for HLA_n nanomotors.

Figure S19. Time-lapse images (Movie S1) displaying the tracking trajectories of nanomotors under (A) HLA₅, (B) HLA₁₀, (C) HLA₁₅, and (D) HLA₂₀ in 15 s (scale bar: 5 μm).

Figure S22. The statistics on speed for HLA_n nanomotors (10 distinct samples, 5 nanomotors were taken from each Movie).

Lastly, there are major sentence structure and grammatical mistakes throughout the manuscript. This causes major difficulties in reading and understanding the text and the results that the authors want to share.

Answer: Thanks so much for the reviewer's kind suggestion. We tried our best to modify our manuscript and made many changes in the revised manuscript, which are marked with red color. Moreover, we also asked Dr. Hong Ying Shen from Massachusetts General Hospital of USA to help us with English writing.

Specific comments

1. Why is there only one particle shown in each Movie? Is the bubble formation because of the reaction, or is it caused by heterogeneous condensation on the surface?

Answer: Thanks a lot for the reviewer's questions.

Question 1:

In fact, we have also taken the Movies with multiple particles for HLA_n nanomotors, and the results were presented and summarized in Movie S1 and Figure S19 of the revised manuscript. It can be seen from the results that when plenty of particles appear in one Movie, large numbers of bubbles were continuously generated, which would greatly influence the study of the movement behavior of a single nanomotor. Hence, in our original manuscript, a single particle was chosen as the study object to investigate its movement behavior. Now, according to the reviewer's suggestion, these Movies with multiple nanomotors were added in the revised manuscript. In particular, the statistics on the speed for HLA_n nanomotors were conducted in order to obtain the speed distribution diagrams of these nanomotors. As shown in Figure S22, the statistical results for HLA₁₀, HLA₁₅, and HLA₂₀ were about 3, 8, and 13 $\mu\text{m s}^{-1}$, respectively.

Figure S19. Time-lapse images (Movie S1) displaying the tracking trajectories of nanomotors under (A) HLA₅, (B) HLA₁₀, (C) HLA₁₅, and (D) HLA₂₀ in 15 s (scale bar: 5 μm).

Figure S22. The statistics on speed for HLA_n nanomotors (10 distinct samples, 5 nanomotors were taken from each Movie).

Question 2:

As illustrated in the literatures, the condensation in supersaturated vapor is a very common phenomenon in atmospheric physics, gas cleaning technology, and multiphase flow. The condensation can be activated *via* two ways including homogeneous nucleation and heterogeneous nucleation. In homogeneous nucleation, a higher nucleation energy barrier should be overcome to create nuclei in the interior of a uniform substance. In heterogeneous nucleation, the foreign surface reduces the nucleation energy barrier and promotes the process of nucleation (*Atmos. Chem. Phys.*, 2009, 9, 1873; *Chem. Eng. Sci.*, 2000, 55, 2895; *J. Chem. Phys.*, 2010, 132, 204504.). By definition, heterogeneous condensation refers to the process of the substance changing from gas containing impurities to a liquid or solid. In our manuscript, gas formation process was discussed, and the bubbles were **continuously generated and disappeared** from nanomotors. Meantime, the produced gas was detected by **using the NO detection kit to verify** that the generated gas was NO. Therefore, the bubble-formation can be attributed to the reaction between L-arginine and H₂O₂ instead of heterogeneous condensation on the nanomotors surface. Meantime, the production **amount of NO during reaction process** was also tested and the results were shown in Figure 3D, further verifying that bubble formation process was owing to reaction.

Figure 3. (A) Time-lapse images (Movie S2) displaying the tracking trajectories of HLA_n nanomotor in 10 s (20% H₂O₂, scale bar: 5 μm), (B) Time-lapse images (Movie S3) displaying the tracking trajectories of HLA₁₀ nanomotor in 10 s under different concentration of H₂O₂ (scale bar: 5 μm), (C) The consumption of L-arginine and (D) production of NO during motion process for HLA₁₀ nanomotor (20% H₂O₂), The speeds of (E) HLA_n nanomotor in 20% H₂O₂ and (F) HLA₁₀ nanomotor in different H₂O₂ concentrations.

2. The concentration of H₂O₂ used during the experiments is much higher than the concentration of reactive oxygen species found in the body (see for example *Xiuzu Song, et. al, Pigm. Cell Melanoma Res., 2009, 809.*). The concentrations of H₂O₂ used for the experiments are toxic and not compatible with in vivo experiments as is suggested by the text.

Answer: Thousands of thanks for the valuable question of the reviewer.

We also thank the reviewer for drawing our attention to this important literature. It is now cited in the revised manuscript.

In our original manuscript, the minimum concentration of the H₂O₂ used in this manuscript under aqueous condition was 5%. In this system, H₂O₂ was used as reactive oxygen species which can be generated by cells in body. As we know, the reactive oxygen, NO synthase (NOS), and other active components can be generated in both blood and cells (*Nat. Rev. Drug Discov., 2008, 7, 156.*). Thus the concentration of H₂O₂ can be decreased in cell-experiment compared with that in aqueous condition (5%). 96-plate was used in the cell-experiment. And the actual cell density in the body is much higher than that in 96-plate performed in the manuscript. Thus the actual reactive oxygen species in real tumor environment may be higher than that in the 96-plate. Hence, extra H₂O₂ was introduced to simulate higher concentration of reactive oxygen species in cells in actual body condition. The choice of concentration (0.2%) was according to a number of literatures (*Angew. Chem. Int. Ed., 2013, 52, 7000; Chem. Phys. Chem., 2014, 15, 2255.*) that investigated the behavior of nanomotors in the cellular environment with the presence of H₂O₂, under which condition the cells can maintain their viability for certain time.

Now, according to the literature suggested by the reviewer (*Pigm. Cell Melanoma Res., 2009, 22, 809.*), the H₂O₂ concentration generated by **melanoma cells** is about 20 pmol/mL/10⁶ cells, which means that the H₂O₂ amount generated by one melanoma cell is about 2*10⁻¹⁷ mol. The cancer cell used in our manuscript is **MCF-7 cell**. According to the literature (*Adv. Mater., 2010, 22, 5164.*), the amount of extracellular H₂O₂ generated by one MCF-7 cell is about 2*10⁻¹³ mol. In general, the number of cells in one 96-well plate for cell experiments is about 10⁵-10⁶ with the cell culture medium volume of 0.2 mL. So we chose 5*10⁵ as the average number of cells. The concentration (mass fraction, %) of extracellular H₂O₂ produced can be calculated by the following formula: C(H₂O₂)=(n*N*M)*100%/V, in which n (mol) represents the amount of H₂O₂ produced by one MCF-7 cell, N representing number of cells, M representing the molar mass of H₂O₂, and V representing the volume of the cell culture medium. The calculated concentration of H₂O₂ by MCF-7 cells in our case was about 0.002%.

According to the questions proposed by the reviewer, we studied the survival rate of cells at different H₂O₂ concentrations. As shown in Figure S23A, MCF-7 cell viability may be greatly influenced by the addition of 0.2% H₂O₂. While MCF-7 cell viability was almost unaffected when the H₂O₂ concentration was 0.002%. So in the cell system of the revised manuscript, 0.002% H₂O₂ was used as extra reactive oxygen species for cell-experiment owing to the fact that the actual cell-density in body was much greater than that in the 96-well plate under experimental conditions, which can better simulate higher concentration of reactive oxygen species in the actual body condition. The choice of concentration (0.002%) was according to the following reasons: (1) The selected concentration of H₂O₂ was similar with the H₂O₂ concentration produced by cells themselves. (2) MCF-7 cell viability was almost unaffected when the concentration of H₂O₂ was 0.002% (Figure S23). The movement behavior of nanomotors in the cellular environment, the uptake behaviors of HLA₁₀ nanomotors by MCF-7 under different conditions (with or without extra H₂O₂ (0.002%)) were also investigated in our revised manuscript.

Confocal laser scanning microscopy was used to observe the movement behavior of nanomotors in the cellular environment, and the results are shown in Figure S24 and Movie S4. Surprisingly, it can be observed that the nanomotors can move both in the MCF-7 cell environment with and without extra-addition of H_2O_2 while the nanomotors displayed only Brownian movement in the pure aqueous environment. Obviously, the movement of nanomotors became much more vigorous in the solution containing 0.002% H_2O_2 and MCF-7. It can also be seen that the nanomotors still maintained a certain vibration after entering the cell due to the existence of the reactive oxygen, NOS, and other active components in the cells.

Besides, the uptake behaviors of HLA_{10} nanomotors by MCF-7 under different conditions (MCF-7+ HLA_{10} nanomotor, MCF-7+0.002% H_2O_2 + HLA_{10} nanomotor, MCF-7+NOS+ HLA_{10} nanomotor, MCF-7+NOS+0.002% H_2O_2 + HLA_{10} nanomotor,) were investigated (Figure 4). As shown in Figure 4, The morphology of MCF-7 was not affected by the addition of 0.002% H_2O_2 . After co-culture with HLA_{10} nanomotors (without extra H_2O_2 addition, 3 h), a small amount of fluorescent particles appeared in the MCF-7 cells. When MCF-7 was incubated with HLA_{10} nanomotors with extra H_2O_2 (0.002%), the amount of fluorescent particles in the cells were enhanced. Moreover, the existence of NOS can also greatly promote the cell uptake of the nanomotors. And MCF-7 cells with both extra H_2O_2 (0.002%) and NOS addition displayed the largest amount of nanomotors in cells. Results indicated that the increase in the concentration of reactive oxygen can promote the movement of the nanomotors, thereby promoting uptake of the nanomotors by the cells. The uptake behaviors of HLA_{10} nanomotors by HUVECs under different conditions also displayed similar trend (Figure 5).

These results indicated that the nanomotor we proposed may move directly under cancer cell environment of experiment, and the addition of 0.002% H_2O_2 can significantly promote the movement and the uptake of nanomotors by the cells. Therefore, we speculated that nanomotors may have the same move ability and uptake behaviour in a real high-density tumor cell environment of body like the condition of MCF-7 cells with extra- H_2O_2 (0.002%).

Figure S23A. Cell viability of MCF-7 co-cultured with different H_2O_2 concentrations.

Figure S24. Time-lapse images (Movie S4) displaying the tracking trajectories of HLA₁₀ nanomotor under (A) PBS solution, (B) MCF-7 cell solution, and (C) MCF-7 cell + 0.002% H₂O₂ in 1 s captured by confocal laser scanning microscopy (scale bar: 5 μm).

Figure 4. Confocal laser scanning microscopy images of the cellular uptake of HLA₁₀ nanomotor by MCF-7 under different conditions: (A) MCF-7+ HLA₁₀ nanomotor, (B) MCF-7+0.002% H₂O₂+HLA₁₀ nanomotor, (C) MCF-7+NOS+HLA₁₀ nanomotor, (D) MCF-7+NOS+0.002% H₂O₂ + HLA₁₀ nanomotor, and (E) a snapshot of a 3D rendered Movie (Movie S5) made from a stack of confocal images (Blue: nanomotors; Green: cell membrane).

Figure 5. Confocal laser scanning microscopy images of the cellular uptake of HLA₁₀ nanomotor by HUVECs under different conditions: (A) HUVECs+HLA₁₀ nanomotor, (B) HUVECs+0.002% H₂O₂+HLA₁₀ nanomotor, (C) HUVECs+NOS+HLA₁₀ nanomotor, (D) HUVECs +NOS+0.002% H₂O₂ + HLA₁₀ nanomotor (E) a snapshot of a 3D rendered Movie (Movie S6) made from a stack of confocal images. Blue: nanomotors; Green: cell membrane.

3. There are some subjective statements made throughout the paper, that require quantification (for example, authors write that the nanomotors have a good fluorescent property thanks to the HPAM component, but do not quantify what is good in this case, or compared to what. In another statement, the consumption of L-arginine is faster and then slower... these statements need to be quantified).

Answer: A lot of thanks for the valuable suggestion of the reviewer.

According to the direction of the reviewer, the fluorescence spectra of HPAM water solution (4 mg mL⁻¹) with different excitation wavelength, HPAM water solution with different concentration, and HLAN nanomotors were detected, respectively. From Figure S14A, the wavelength of 420 nm was chosen as the excitation wavelength for subsequent experiments. It can be seen from Figure S14B, the fluorescence intensity increased with the increasing concentration of HPAM, and the fluorescence intensity change between 80-700 (a.u.), indicating good fluorescent property of HPAM. Meantime, the fluorescence intensities of HLAN nanomotors were about 60 (Figure S14C), which can provide cell imaging ability for the nanomotors.

Moreover, the consumption rate of L-arginine was calculated and the relevant revised statement was added into the revised manuscript: “The consumption rate of L-arginine was about 17.20 μM/min for the first three hours, which decreased to 0.03 μM/min and 0.02 μM/min for the following 3-6 h and 6-9 h, respectively.”

Figure S14. Fluorescence spectra of (A) HPAM water solution (4 mg mL^{-1}) with different excitation wavelengths; (B) HPAM water solution with different concentrations (excitation wavelength: 400 nm); and (C) HLA_n nanomotors.

4. Also in the Movies. The captions of the different particles that are shown states that these are HLA_n motors ($n=5,10$, and so on). I don't understand why is it possible to observe them with a light microscope. Why would polymers scatter so much, to make them visible under a light microscope? It seems more likely that the Movies show some aggregates of the particles than just a single one as suggested by the captions.

Answer: Thanks a lot for the helpful comments from the reviewer.

In general, the size of micro- or nanomotors in most literatures is micron ($1\text{-}50 \mu\text{m}$) (*ACS Nano*, 2016, 10, 10389; *Adv. Funct. Mater.*, 2015, 25, 7497; *Nano Lett.*, 2016, 16, 2860; *J. Am. Chem. Soc.*, 2016, 138, 6492; *ACS Nano*, 2016, 10, 9536.), so the individual particles with micrometer size can be easily observed with light microscope. According to the literatures, the nanomotors with small size ($<200 \text{ nm}$) can hardly be observed by normal optical microscopes owing to the relative lower resolution of normal optical microscopes (*Chem. Phys. Chem.*, 2014, 15, 2255; *J. Am. Chem. Soc.*, 2016, 138, 6492.). The single particle of nanomotors (HLA₅ and HLA₁₀) prepared in our case was difficult to observe under light microscope due to their small size (less than 200 nm) and good dispersibility (Figure R1). As seen in Figure R2, only some continual bubbles can be observed, and the bubbles continue to produce after disappearing, but HLA₅ nanoparticles cannot be observed in light microscope due to their small size. Generally, when we observe the movement behavior of small-sized nanomotors, **we mainly focus on the bubbles generated by the nanomotors rather than the nanomotors themselves owing to their small size.**

According to literature (*J. Am. Chem. Soc.*, 2016, 138, 6492.), confocal laser scanning microscopy can be used to investigate the movement behavior of nanomotors with small size. Hence in our revised manuscript, confocal laser scanning microscopy was used to observe the movement behavior of nanomotors in the cellular environment, and the results were shown in Figure S24 and Movie S4. It can be observed that the nanomotors can move both in the MCF-7 cell environment with and without extra-addition of H_2O_2 while the nanomotors displayed only Brownian movement in the pure aqueous environment. Obviously, the movement of nanomotors became much more vigorous in the solution containing 0.002% H_2O_2 and MCF-7. It can also be seen that the nanomotors still maintained a certain vibration after entering the cell due to the existence of the reactive oxygen, NOS, and other active components in the cells.

We speculated that the reason for the aggregation of nanomotors may be that the nanomotors we prepared were difficult to observe under light microscope due to their small size. Therefore, a higher concentration (about 10 mg/mL) of the nanomotors was used when preparing samples for Movie

capturing, which may cause aggregation phenomena of the nanomotors. Meantime, we have also taken the Movies with multiple particles for HLA_n nanomotors under relative low concentration of nanomotors (about 1 mg/mL), and the results were presented and summarized in Movie S1 and Figure S19 of the revised manuscript. It can be seen from the results that when plenty of particles appear in one Movie, large numbers of bubbles were continuously generated, which would greatly influence the study of the movement behavior of a single nanomotor. Hence, in our original manuscript, a single particle was chosen as the study object to investigate its movement behavior. Moreover, the statistics on the speed for HLA_n nanomotors were conducted in order to obtain the speed distribution diagrams of these nanomotors. As shown in Figure S22, the statistical results for HLA₁₀, HLA₁₅, and HLA₂₀ were about 3, 8, and 13 $\mu\text{m s}^{-1}$, respectively.

Thanks again for the kind reminding of the reviewer, we added this part in our revised manuscript and marked it with red color.

Figure R1. Photographs of the (A) HLA₅ and (B) HLA₁₀ dispersed in water.

Figure R2. Time-lapse images displaying the bubble generation process of HLA₅. The scale bar is 50 μm .

5. The TEM characterization of the particles is not done properly. The images shown in figure 2 are not properly focused and a lot of information is missing to back up the claims made in the paper.

Answer: Many thanks for the valuable question of the reviewer.

According to many literatures (*Nanoscale*, 2014, 6, 2730; *Nature*, 2015, 526, 118; *Adv. Mater.*, 2016, 28, 9581.), some suitable negative stains include ammonium molybdate, uranyl acetate, uranyl formate and phosphotungstic acid can be used for better observation of the polymeric nanoparticles because they scatter electrons strongly and also adsorb to biological matter well. In this technique, the background is stained, leaving the actual specimen untouched, and thus visible. In order to characterize the morphologies and particle sizes of the nanomotors we prepared, the nanomotors were first negatively stained with uranyl acetate and then observed using transmission electron microscopy (TEM) (Figure S1). The resulting image revealed spherical morphology of the nanomotors and the particle sizes for HLA₅, HLA₁₀, and HLA₁₅ matched the hydrodynamic diameter variation trend measured by dynamic light scattering (DLS, Figure S2). For HLA₂₀ sample, the particle size was irregular, and obvious crystallization phenomenon appeared (Figure S3). The particles of HLA₂₀ can be easily

observed by TEM, hence TEM image of HLA₂₀ without being negatively stained was obtained (Figure S3), which showed that particle size of HLA₂₀ was larger than 1 μm .

Thanks again for the kind reminding of the reviewer, we added this part in our revised manuscript and marked it with red color.

Figure S1. TEM images of (A) HLA₅, (B) HLA₁₀, and (C) HLA₁₅ negatively stained with uranyl acetate.

Figure S2. Particle size distribution of HLA₅, HLA₁₀, and HLA₁₅ nanomotors.

Figure S3. TEM image of HLA₂₀.

6. For the TEM analysis, there is also no statistical analysis of multiple particles, at least one image (could be in the SI) should show an overview of the sample. The HLA₂₀ image does not even show where the particle “ends”. In the comparison of the particles being destroyed (Fig. 2C (a-e)), the size of the images should be consistent to make it easier for a real comparison.

Answer: Many thanks for the kind suggestion of the reviewer.

According to the reviewer’s suggestion, the statistical analyses of multiple particles were conducted by several TEM images. As shown in Figure S4, the obtained particle sizes after statistical analyses for HLA₅, HLA₁₀, and HLA₁₅ were about 130, 160, and 390 nm, respectively, which matched the particle sizes shown in Figure 1 and also matched the variation trend of hydrodynamic diameter measured by dynamic light scattering (DLS, Figure S2).

For HLA₂₀ sample, the particle size was irregular, and obvious crystallization phenomenon appeared. The particles of HLA₂₀ can be easily observed by TEM (Figure S3). The complete image of HLA₂₀ was shown in Figure S3, which showed that particle size of HLA₂₀ was larger than 1 μm .

Besides, to make it easier for a real comparison of the nanomotors being destroyed, the revised images of Fig. 2C (a-e) was as follows:

Figure 2. (A) TEM images of (a) HLA₅ nanomotor, (b) HLA₁₀ nanomotor, (c) HLA₁₅ nanomotor and (d) HLA₂₀ nanomotor, (B) MS spectra of (a) L- arginine, (b) HPAM, (c) L-citrulline, (d) HLA₁₀ nanomotor, and (e) HLC (supernatant HPAM/L-citrulline composite after HLA₁₀ nanomotor reacting with 10% H₂O₂ for 24 h), (C) Self-destroyed process of HLA₁₀ nanomotor in 10% H₂O₂ for (a) 0 h, (b) 1 h, (c) 3 h, (d) 6 h, and (e) 18 h, respectively.

Figure S4. The statistics on particle size (obtained from TEM images, five distinct samples, 10 particles were taken from each image) for HLA_n nanomotors.

7. Coming back to the proposed mechanism for the directionality of the bubble formation, how do the authors explain a migration of the reactions products, to one specific site of the particle? Why only to one specific site? In addition, this conclusion is contradictory to the images seen in the supporting Movie S1 and supporting information S8, where the HLA₁₅ structures show 2 bubbles forming on opposite sites.

Answer: Thanks so much for the question of the reviewer.

In order to interpret the migration of the reaction products, the mechanism for the directionality of

the bubble formation was modified and completed as follows:

Based on the preparation method, L-arginine was evenly distributed on the outside of the HLA₁₀ nanomotors, so the small bubbles can be generated in all sites of the nanomotor. These small bubbles would quickly aggregate to form a large bubble due to the small size of the nanomotor. Therefore, it can be seen from the movie that the bubble gradually aggregated and increased with reaction time. When the bubble grew on the nanomotor surface, the nanomotor would move away from the center of the bubble owing to the bubble growth force. The bubble would suddenly disappear once it reached a maximum radius (*Phys. Rev. Letters*, 2012, 109, 128305.), then the nanomotor can be driven after the pressure in large bubbles being high enough to overcome the surface energy of the gas-liquid interface. The nanomotors can continually generate small bubbles, and these small bubbles were subject to the friction of the moving nanomotor. The direction of the friction was opposite to the direction of movement of the nanomotor. As shown in Figure S20, the small bubbles accumulated in the direction of the friction to the large bubbles and continued to grow into large bubbles, thus continuing to push the nanomotors. And the subsequent bubble generation would grow in this direction. As a result, when the concentration of L-arginine was low (HLA₅), the generation rate (0.023 bubble/s, Figure S21) of the bubbles was too small to promote its movement during the process of growing up. The rate of bubble generation (about 0.5 bubble/s for HLA₁₀) increased with the amount of L-arginine, which can facilitate the movement of the nanomotors. As for HLA₁₅ and HLA₂₀, the rates of bubble generation were much faster (0.6 and 0.8 bubble/s) owing to the high amount of L-arginine in the nanomotors. Since the bubble generation speeds of these two nanomotors (HLA₁₅ and HLA₂₀) were much faster, the bubble density on the nanomotors surface were much higher, so the small bubble would rapidly grow into a large bubble, and more than one large bubbles may be generated due to high density of small bubbles (*J. Am. Chem. Soc.*, 2018, 140, 11902.), so the motion direction of the nanomotor was related to the direction of the resultant force. As shown in Figure S20, the motion direction of the nanomotor was not a straight line but a curve (Figure 3A).

Based on this mechanism, we speculated that the migration of the product may be in two ways. One of the products was L-citrulline, which was produced uniformly around the nanoparticles and gradually diffuses into the surrounding solution. Another product of NO was released to form bubbles, which were uniformly generated around the nanomotors, and then gradually grew up to overcome the surface energy between gas and liquid. Therefore, the direction of migration of NO may be mainly the opposite direction of motion. The mechanisms of bubble generation and motion of nanomotors are very complicated (*Chem. Phys. Chem.*, 2014, 15, 2255; *J. Am. Chem. Soc.*, 2016, 138, 6492; *J. Am. Chem. Soc.*, 2018, 140, 11902.), and more research will continue in future.

Thanks again for the kind reminding of the reviewer, we added this part in our revised manuscript and marked it with red color.

Figure S20. Possible motion mechanism of (A) the HLA₁₀ and (B) HLA₁₅/HLA₂₀ nanomotors during bubble growth process.

Figure S21. Time-lapse images (Movie S2) displaying the bubble generation process of (A) HLA₅, (B) HLA₁₀, (C) HLA₁₅, (D) HLA₂₀ in 10 s, respectively (20% H₂O₂).

Figure 3. (A) Time-lapse images (Movie S2) displaying the tracking trajectories of HLA_n nanomotor in 10 s (20% H₂O₂, scale bar: 5 μm), (B) Time-lapse images (Movie S3) displaying the tracking trajectories of HLA₁₀ nanomotor in 10 s under different concentration of H₂O₂ (scale bar: 5 μm), (C) The consumption of L-arginine and (D) production of NO during motion process for HLA₁₀ nanomotor (20% H₂O₂), The speeds of (E) HLA_n nanomotor in 20% H₂O₂ and (F) HLA₁₀ nanomotor in different H₂O₂ concentrations.

8. The figures are key to understanding the entire article and the recognition of many symbols in the images is hard or even impossible. It would be necessary to improve the images before any publication is possible. This includes the fluorescent images.

Answer: Thanks so much for the kind suggestion of the reviewer.

According to the advice of the reviewer, we revised the figures listed in our revised manuscript. Meantime, in order to enhance the quality of the fluorescent images, the confocal laser scanning microscopy and 3D rendered Movies made from a stack of confocal images were used to investigate cell uptake behavior of nanomotors. The results were shown in Figure 4 and 5 of the revised manuscript.

9. From the images presented for the fluorescence studies, it is impossible to determine if the nanomotors are inside or on the surface of the cells. Being the particles made out of polymers, it is likely that they would be attracted to the cellular membrane, so how can the authors know this is not the case.

Answer: Thanks so much for the valuable question of the reviewer.

In order to determine whether the nanomotors are inside or on the surface of the cells, the confocal laser scanning microscopy and corresponding 3D reconstruction images (Movies) were further used to characterize the location of these nanomotors (*ACS Nano*, 2018, 12, 10212.). Microscope images were taken under a 100x magnification oil objective using confocal laser scanning microscopy. In addition to capturing normal images, Z-stacks were recorded to obtain an orthogonal view of the cells and 3D images, in which green color represented cell membrane and blue color represented the nanomotors (Figure 4E and 5E). By cutting the green channel of the 3D images, it can be observed that the green cell membrane was gradually peeled off, exposing the internal blue color (HLA₁₀), confirming the intracellular localization of the HLA₁₀ nanomotors. Results of Figure 4E and 5E can confirm the intracellular localization of the nanomotors. Thanks again for the kind reminding of the reviewer, we added this part in our revised manuscript and marked it with red color.

10. The results presented in the SI seem not ready for publication. Images that overlap each other, wrong label of in the graphs, different thickness of plot lines given for comparison.

Answer: We thank the reviewer for his/her careful work to help us improving the manuscript and apologize for our negligence. According to the advice of the reviewer, we have tried our best to revise the figures listed in supporting information. Thanks again for the kind reminding of the reviewer.

11. The Supplementary information order is inappropriate and there is no reference to S9 in the main text.

Answer: We really appreciate the reviewer's kind reminding and apologize for our negligence. In our revised manuscript, Figure S9 (Figure 15 in the revised manuscript) was added in the main text.

12. The list of references is focused on very few contributors to the field of nanomotors. Although I do not doubt, that these groups are great contributors to the field, the authors might consider other key literature from different authors. As a suggestion, authors might consider work published by the groups of D. Wilson (*ACS Nano*, 2017, 11, 1957.), Q. He (*J. Am. Chem. Soc.*, DOI: 10.1021/jacs.8b06646), P. Fischer (*Nano Lett.*, 2014, 14, 2407.), B. Staedler (*Chem. Mater.*, 2015, 27, 7412.), A. Sen (*Angew. Chem. Int. Ed.*, 2011, 50, 9374.) among many others. Most importantly, there are many mistakes in the

citations (name, surname order does not match between citations).

Answer: We thank the reviewer for drawing our attention to these important articles, which have been cited in our revised manuscript. Meantime, we have tried our best to check the name and surname order in our revised manuscript and marked with red color. Literatures from D. Wilson (*ACS Nano*, 2017, 11, 1957.), Q. He (*J. Am. Chem. Soc.*, DOI: 10.1021/jacs.8b06646), P. Fischer (*Nano Lett.*, 2014, 14, 2407.), B. Staedler (*Chem. Mater.*, 2015, 27, 7412.), and A. Sen (*Angew. Chem. Int. Ed.*, 2011, 50, 9374.) were referred as reference 9, 32, 12, 10 and 47, respectively. Thanks again for the kind reminding of the reviewer.

Further notes

1. I'm not an expert, but as far as my understanding goes, HPAM will degrade to produce acrylamide, a very toxic material. Since this is the material which the nanomotors are made of, is this not enough reason to think that the particles here presented are also very toxic?

Answer: Many thanks for the helpful comments from the reviewer.

Dendritic molecules, such as polyamidoamine, polylysine, polyester, polyglycerol (PG), and triazine dendrimers, have been introduced for biomedical applications to amplify or multiply molecularly pathopharmacological effects. Among which, Hyperbrached Polyamines (HPAM) dendrimer molecules have been demonstrated considerable efficacy in both gene therapy and drug delivery owing to their high level of transfection in a wide variety of cells in culture with low cytotoxicity (*Polym. J.*, 1985, 17, 117; *Macromolecules*, 1986, 19, 2466; *J. Control. Release.*, 2004, 99, 445; *Drug Deliv.*, 2016, 23, 2956; *Biomaterials*, 2009, 30, 6976; *Biomaterials*, 2009, 30, 6109.).

In general, the synthetic monomers of HPAM includes many types, and the typical two groups of monomers are as follows: N,N'-cystaminebisacrylamide and 1-(2-aminoethyl) piperazine (Scheme R1), methacrylate and ethylenediamine (Scheme R2).

For the HPAM containing disulfide linkages prepared via Michael addition polymerization of N,N'-cystaminebisacrylamide and 1-(2-aminoethyl) piperazine (Scheme R1) (*Biomacromolecules*, 2010, 11, 1840.), good biodegradable property can be obtained owing to the fact that abundant S-S bonds in their backbones can be easily cleaved in the presence of biological or chemical stimuli (*Biomacromolecules*, 2011, 12, 1523; *Bioconjugate Chem.*, 2007, 18, 138; *J. Am. Chem. Soc.*, 2007, 129, 5354.).

For the HPAM synthesized by using methacrylate and ethylenediamine as monomers, also showed very good biocompatibility (Scheme R2) (*Bioconjugate Chem.*, 1996, 7, 703.), and degradation is rather difficult to achieve owing to its stable property. Transfection of cultured cells has been reported using complexes between DNA and spherical cationic HPAM of this kind that consist of primary amines on the surface and tertiary amines in the interior. Some researchers reported that heat treatment in a variety of solvolytic solvents, e.g., water or butanol can induce significant degradation of the dendrimer at the amide linkage, resulting in a heterodisperse population of compounds with much lower molecular weights. It has been suggested that the degradation process occurs by solvolysis of peptide bonds in the dendrimer. Surprisingly, the transfection activity of the dendrimers is dramatically enhanced (>50-fold) after random degradation by heat treatment in n-butanol/H₂O (*Langmuir*, 2007, 23, 737; *Bioconjugate Chem.*, 1996, 7, 703; *Biomacromolecules*, 2009, 11, 245; *Biomacromolecules*, 2009, 10, 2921; *J. Control. Release*, 2008, 126, 59.). The above results illustrate two points: First, HPAM is difficult to degrade under common human environment, requiring use of organic solvent and heating condition. Second, even if HPAM synthesized by using methacrylate and ethylenediamine as

monomers is degraded, the degradation process involves the cleavage of amide linkage instead of going back to its synthetic monomer, causing no biological toxicity.

Moreover, the fact that HPAM is difficult to degrade under common human environment can also be proved by the difficulty of degradation of polyamide. Researchers had verified that polyamide can degrade in chlorinated water or by heating condition. According to previous studies, N-chlorination is certainly one reaction that ultimately contributes to polyamide degradation, involving Cl_2 or HClO (*Desalination*, 1994, 95, 325; *Helv. Chim. Acta*, 2001, 84, 2549.).

The HPAM used in this paper is synthesized by using methacrylate and ethylenediamine as monomers. It has very good biocompatibility and stability because it is hardly degrade in the human environment. Besides, the HPAM can be removed naturally by human body through glomerular filtration, since its molecular weight is lower than 15000. (*Clin. Chim. Acta*, 2000, 297, 55.)

Thanks again for the kind reminding of the reviewer, we added this part in our revised manuscript and marked it with red color.

Scheme R1. Synthetic process of HPAM with N, N'-cystaminebisacrylamide and 1-(2-aminoethyl) piperazine as monomers (*Biomacromolecules*, 2010, 11, 1840).

Scheme R2. General synthesis scheme of HPAM with methacrylate and ethylenediamine as monomers (*Bioconjugate Chem.*, 1996, 7, 703.).

2. To check that the formation of the particles happens as stated, one suggestion would be to modify the surface of the structures by the exposed (expected) group. This way the authors can prove the suggested formation mechanism.

Answer: Many thanks for the helpful suggestion from the reviewer.

The possible binding mechanism of HLA_n proposed in our case was that the $-\text{COOH}$ group in L-arginine being attracted by positive $-\text{NH}_2$ group, leaving $-\text{C}=\text{NH}$ and $-\text{NH}_2$ groups of L-arginine locating outside of the HLA_n nanomotors and retaining high reactive functional group ($-\text{C}=\text{NH}$) of L-arginine. In order to verify the exposed functional group of HLA_n samples, FITC (Fluorescein isothiocyanate isomer I, (Figure S11), which can react with $-\text{NH}_2$ group to form covalent bond), was used to modify the surface of HLA₁₀ to form FITC-HLA₁₀. The fluorescence spectra of FITC, HLA₁₀,

and FITC-HLA₁₀ were detected to characterize whether FITC can be modified on the surface of HLA₁₀. As shown in Figure S12, FITC-HLA₁₀ displayed similar fluorescence spectrum (peak located at 510 nm) as FITC with slightly decreased intensity, while HLA₁₀ displayed no peak at the wavelength of about 510 nm. These results verified the fact that the exposed functional groups of HLA₁₀ nanomotors were -NH₂ groups, proving the suggested formation mechanism.

Meantime, the highly negative charged molecule heparin (structure of heparin was shown in Figure S13) was also used to react with L-arginine to further prove the proposed formation mechanism. According to the proposed mechanism, the negative charged heparin can form nanoparticles with L-arginine (denoted as Hep/L-arginine) through electrostatic attraction between -COO⁻ from heparin and the -NH₂ groups from L-arginine. Hence, -NH₂ groups in L-arginine were covered by -COO⁻ from heparin. As a result, FITC cannot react with Hep/L-arginine. As shown in Figure S12, neither Hep/L-arginine nor FITC-Hep/L-arginine showed peaks at 510 nm, indicating that FITC cannot react with Hep/L-arginine, further verifying the proposed formation mechanism of HLA₁₀ nanomotors.

Thanks again for the kind reminding of the reviewer, we added this part in our revised manuscript and marked it with red color.

Figure S11. Chemical structure of FITC.

Figure S12. Fluorescence spectra of different samples (Excitation wavelength: 490 nm).

Figure S13. Chemical structure of heparin.

Reviewer #2 (Remarks to the Author):

Authors developed here for the first time a new type of zero-waste and self-destroyed HLA nanomotors utilizing L-arginine as fuel, NO as driving force and L-citrulline as a beneficial by-product. Results presented demonstrated the potential of these HLA nanomotors as self-imaging nanomotors (due to their good fluorescence performance) and for endothelialization and anticancer effect (taking advantage of the use of NO both as the driving force and to promote angiogenesis and kill cancer cells). Authors perform an exhaustive characterization of the developed nanomotors by TEM, MS, Zeta potential, FTIR and XPS and their motion performance using different L-arginine and H₂O₂ concentrations. Results presented demonstrated also the uptake of HLA nanomotors by HUVECs, the possibility to kill cancer cells, due to the higher amount of NO produced by the HLA nanomotors as a result of the higher basal level concentration of reactive oxygen species in these cells, and the possibility to extend the protocol to synthesize other kinds of nanomotors expelled by NO using different amino-enriched organic compounds. Although the results presented are interesting, my major concern about this paper is to clarify the advantages of these NO-expelled artificial nanomotors compared to (US)-powered nanowire motors, which have demonstrated excellent properties for internalization, killing selectively cancer cells within 5 min (Ref 3 in the paper) and even single cell real-time using attractive intracellular “OFF-ON” fluorescence switching mechanisms (*ACS Nano*, 2015, 9, 6756.). Other concerns that should be addressed before recommending publication of this paper in this top Journal:

Answer: We thank the reviewer so much for his/her comment on the novelty of our work. We also thank the reviewer so much for offering these two important articles, which have been cited in the revised manuscript.

It is quite true that ultrasound (US)-powered Au nanowire (or graphene oxide modified Au nanowires) motors loaded with caspase-3 (CASP-3) demonstrated excellent properties for internalization, killing selectively cancer cells within 5 min (*ACS Nano*, 2017, 11, 5367.) and even single cell real-time using attractive intracellular “OFF-ON” fluorescence switching mechanisms (*ACS Nano*, 2015, 9, 6756.). Compared with physical micro/nanomotors (e.g. (US)-powered nanowire motors), the advantages of the NO-expelled artificial nanomotors we proposed can be summarized as follows:

(1) Unlike physical micro/nanomotors which are propelled by magnetic field, ultrasound or light often relied on some complicated actuation system to maintain their motion due to their lack of self-driven ability (*J. Am. Chem. Soc.*, 2016, 138, 6492; *Adv. Mater.*, 2017, 29, 1604825; *Mater. Horiz.*, 2016, 3, 113; *Adv. Mater.*, 2018, 1800429.). Hence, the concept of chemical self-driving force offers more selectivity and possibilities for preparation and application of micro/nanomotors.

(2) Compared with these nanomotors generally loaded with anticancer drugs (such as Cisplatin, DOX, caspase-3, and so on), the anticancer agent used in our case is the continuously generated NO during the movement process of the nanomotors. In general, anticancer drugs can kill cancer cells with very fast rate, yet, may simultaneously cause toxicity to non-tumor cells, resulting in different degrees of damage to normal cells of tissues and organs even if low concentrations of drugs remain (*J. Ovarian Res.*, 2015, 8, 20; *Cancers*, 2011, 3, 1351; *Cancer Cell*, 2015, 28, 690; *Can. J. Cardiol.* 2016,

32, 852.). In comparison, higher NO concentration can cause cancer cells to die, and the residual NO concentration after killing cancer cells will be gradually decreased by the dilution of blood, which is associated with several beneficial functions such as cognitive function, regulating the non-adrenergic/non-cholinergic relaxation of smooth muscle cells and acting as a therapeutic agent promoting angiogenesis (*Nat. Rev. Neurosci.*, 2007, 8, 766.). Not only that, another product (L-citrulline) during the movement process of the nanomotors we proposed can improve immune system function, maintain joint function, balance normal blood sugar levels, contain rich antioxidants absorbing harmful free radicals (*Angew. Chem. Int. Ed.*, 2017, 56, 1229; *Int. J. Radiat. Oncol.*, 2003, 57, 1067.). Thus, the nanomotors we proposed in this manuscript is a truly zero-waste nanomotors, in which both the reactant and the products can have a beneficial effect on human body.

(3) The concept of self-destroyed design is another advantage of the NO-propelled nanomotors in our case compared with physical micro/nanomotors (e.g. (US)-powered nanowire motors). As illustrated by many researchers (*ACS Nano*, 2016, 10, 10389.), self-destroyed nanomotors became one of the ideal choice for that they will disappear after completing their task without causing extra damage to human body (*Nature*, 2017, 545, 406.). Micro/nanosized insoluble residues (such as Au nanowires and graphene oxide) may behave potential hazard (*Science*, 2006, 311, 622.). Moreover, they may readily travel throughout the human body to accumulate in vital organs and trigger serious immune responses due to unintended interactions at the cellular, subcellular, or protein levels (*Science*, 2006, 311, 622; *Expert Opin. Drug Metab. Toxicol.*, 2012, 8, 47.). The self-destroyed process of the nanomotors in this manuscript was detected by TEM technique (Figure 2C), which illustrated that the particle size was decreased from about 150 nm to about 70 nm for 18 h owing to the weakened interactions between HPAM and L-arginine because of the charge shielding induced by its external environment (*ACS Nano*, 2016, 10, 10389.).

According to the advice of the reviewer, we added this part in “**Results and Discussion**” of our revised manuscript.

Figure 2. (A) TEM images of (a) HLA₅ nanomotor, (b) HLA₁₀ nanomotor, (c) HLA₁₅ nanomotor and (d) HLA₂₀ nanomotor, (B) MS spectra of (a) L- arginine, (b) HPAM, (c) L-citrulline, (d) HLA₁₀ nanomotor, and (e) HLC (supernatant HPAM/L-citrulline composite after HLA₁₀ nanomotor reacting with 10% H₂O₂ for 24 h), (C) Self-destroyed process of HLA₁₀ nanomotor in 10% H₂O₂ for (a) 0 h, (b) 1 h, (c) 3 h, (d) 6 h, and (e) 18 h, respectively.

1. The viability of HUVECs after HLA nanomotors intake (incubation of HUVECs with HLA nanomotors under 0.2% H₂O₂) should be indicated at different times.

Answer: We thank the reviewer a lot for his/her valuable suggestions.

In our original manuscript, the minimum concentration of the H₂O₂ used in this manuscript under aqueous condition was 5%. In this system, H₂O₂ was used to mimic reactive oxygen species which can be generated by cells in body. As we know, the reactive oxygen, NOS, and other active components can be generated in both blood and cells (*Nat. Rev. Drug Discov.*, 2008, 7, 156.). Thus the concentration of H₂O₂ can be decreased in cell-experiment compared with that in aqueous condition (5%). 96-plate was used in the cell-experiment. And the actual cell density in the body is much higher than that in 96-plate performed in the manuscript. Thus the actual reactive oxygen species in real tumor environment may be higher than that in the 96-plate. Hence, extra H₂O₂ was introduced to simulate higher concentration of reactive oxygen species in cells in actual body condition. The choice of concentration (0.2%) was according to a number of literatures (*Angew. Chem. Int. Ed.* 2013, 52, 7000; *ChemPhysChem* 2014, 15, 2255.) that investigated the behavior of nanomotors in the cellular environment with the presence of H₂O₂, under which condition the cells can maintain their viability for certain time.

The cancer cell used in our manuscript is **MCF-7 cell**. According to the literature (*Adv. Mater.* 2010, 22, 5164), the amount of extracellular H₂O₂ generated by one MCF-7 cell is about 2×10^{-13} mol. In general, the number of cells in one 96-well plate for cell experiments is about 10^5 - 10^6 with the cell culture medium volume of 0.2 mL. So we choose 5×10^5 as the average number of cells. The concentration (mass fraction, %) of extracellular H₂O₂ produced can be calculated by the following formula: $C(\text{H}_2\text{O}_2) = (n \cdot N \cdot M) \cdot 100\% / V$, in which n (mol) represents the amount of H₂O₂ produced by one MCF-7 cell, N representing number of cells, M representing the molar mass of H₂O₂, and V

representing the volume of the cell culture medium. The calculated concentration of H_2O_2 by MCF-7 cells in our case was about 0.002%.

According to the questions proposed by the reviewer, we studied the survival rate of cells at different concentrations of H_2O_2 . As shown in Figure S23, the viability of MCF-7 cells and HUVECs may be greatly influenced by the addition of 0.2% H_2O_2 , yet cell viability was almost unaffected when the concentration of H_2O_2 was 0.002%. So in the cell system of the revised manuscript, 0.002% H_2O_2 was used as extra reactive oxygen species for cell-experiment owing to the fact that the actual cell-density in body was much greater than that in the 96-well plate under experimental conditions, which can better simulate higher concentration of reactive oxygen species in the actual body condition.

According to the reviewer's advice, the viabilities of HUVECs after HLA nanomotors intake (incubation of HUVECs with HLA nanomotors under 0.002% H_2O_2) at different times were detected by MTT method. As shown in Figure S25, the cell viability of HUVECs after HLA nanomotors intake gradually increased with the cultured time (1-7 d).

Thanks again for the kind reminding of the reviewer, we added this part in our revised manuscript and marked it with red color.

Figure S23. Cell viabilities of (A) MCF-7 and (B) HUVECs co-cultured with H_2O_2 under different concentrations.

Figure S25. MTT results of HUVECs after cell-uptake process with HLA₁₀ nanomotors under 0.002% H₂O₂ for different times.

2. This phrase “Especially, the amount of NO produced in MCF-7 (265 μM) was significantly higher than that in HUVECs (85 μM) with the 150 μL of the added HLA10 (4 h)” given in lines 238-240 required further discussion, such as... the amount of NO in cancer cells is higher because of a more efficient movement of the HLA due to the higher concentration of reactive oxygen species in these cells.

Answer: We thank the reviewer a lot for this valuable advice.

According to the suggestion of the reviewer, we rewrite this part in our revised manuscript as follows: “Especially, the amount of NO produced in MCF-7 (265 μM) was significantly higher than that in HUVECs (85 μM) with the 150 μL of the added HLA10 (4 h). The amount of NO produced in cancer cells was higher because of a more efficient movement of the HLA₁₀ nanomotors due to the higher concentration of reactive oxygen species in these MCF-7 cells than that in HUVECs, in which one MCF-7 cell can produce 2×10^{-13} mol of H₂O₂ (*Adv. Mater.*, 2010, 22, 5164).

3. At present, the selectivity of these HLA to destroy cancer cells is given only by the higher basal level concentration of reactive oxygen species in the cell, authors should discuss with more detail this selectivity issue and ways to improve it, would it be feasible to functionalise these HLAs with adequate bioreceptors for targeted cell killing?

Answer: We thank the reviewer a lot for this insightful question.

It is quite true that the selectivity of these HLA_n nanomotors to destroy cancer cells is given by the higher basal level concentration of reactive oxygen species in the cell. As we know, folic acid receptors (FRs) exhibit limited expression on healthy cells but are often present in large numbers on surface of cancer cells (*Proc. Natl. Acad. Sci. USA*, 1986, 83, 5983.). Thus, FRs represent an important target for tumor-specific delivery of anticancer drugs. Therefore, folic acid (FA) was regarded as an important molecule for FR-mediated targeted delivery for anticancer drugs. In our case, heparin was used to react with folic acid to form HF (Heparin/Folic acid) NPs (*Biomacromolecules*, 2010, 11, 3531.), then HF NPs were used to form HFLA₁₀ NPs (heparin/folic acid/L-arginine) with L-arginine. As shown in Figure 7C, the particle size of HFLA₁₀ NPs was about 500 nm. In order to confirm the targeting effect of HFLA₁₀ NPs for MCF-7 cells, MTT and cell uptake behavior of HLA₁₀ and HFLA₁₀

nanomotors co-cultured with MCF-7 cells were conducted. As shown in Figure S31, the cell viability of MCF-7 cells decreased to about 30% after co-cultured with HFLA₁₀ for 3 h, which was about 41% for HLA₁₀. Furthermore, the cell uptake tests of HLA₁₀ and HFLA₁₀ nanomotors after co-cultured with MCF-7 cells for 3 h were also detected by confocal laser scanning microscopy. As shown in Figure S32, HLA₁₀ and HFLA₁₀ nanomotors displayed similar fluorescence properties, hence the fluorescence intensity detected by confocal laser scanning microscopy can represent the cell uptake amount of nanomotors. As shown in Figure S33, the fluorescence intensity of the MCF-7 cell co-cultured with HLA₁₀ was much lower than that of the MCF-7 cell co-cultured with HFLA₁₀ nanomotors, indicating good targeting effect of HFLA₁₀ nanomotors for MCF-7 cells. So it will be feasible to functionalise these HLAs with adequate bioreceptors for targeted cell killing.

Thanks again for the kind reminding of the reviewer, we added this part in our revised manuscript and marked it with red color.

Figure S31. MTT results of different samples after co-culturing with MCF-7 for 3 h.

Figure S32. Fluorescence spectra of HLA₁₀ and HFLA₁₀.

Figure S33. Confocal laser scanning microscopy images of the cellular uptake of (A) HLA₁₀ and (B) HFLA₁₀ nanomotorS by MCF-7 cells and (D) a snapshot of a 3D rendered Movie (Movie S5) made from a stack of confocal images. Blue: nanomotors; Green: cell membrane.

4. The storage and operational stability and the reproducibility in the fabrication protocol of these new HLA nanomotors should be described in more detail.

Answer: Thanks so much for the reviewer's kind suggestion.

According to the reviewer's advice, the storage and operational stability and the reproducibility in the fabrication protocol of these new HLA_n nanomotors were detected. The synthesized HLA₁₀ nanomotors can be kept for at least 1 month (at room temperature) without sedimentation (Figure S16) and DLS results displayed that the size of HLA₁₀ changed a little (Figure S17), indicating good storage stability of the fabrication protocol. Moreover, the synthesis method was repeated for five times, and DLS of HLA₁₀ was detected to verify good operational stability and reproducibility of the fabrication protocol. As shown in Figure S18, DLS results for the HLA₁₀ nanomotors prepared by five repeated times of experiment were similar with each other, implying good operational stability and reproducibility of the fabrication protocol we used.

Figure S16. Photographs of the HLA₁₀ dispersed in water (A) before and (B) after one-month storage.

Figure S17. Particle size of HLA₁₀ dispersed in water (A) before and (B) after one-month storage (at room temperature).

Figure S18. Particle size of HLA₁₀ prepared by five repeated times of experiment.

5. A minor concern: There are some misspelled words and missing spaces through the text, please revise and address.

Answer: Thanks so much for the reviewer's kind suggestion. We have tried our best to improve our manuscript and made many changes in the revised manuscript, which are marked with red color. Moreover, we also asked Dr. Hong Ying Shen from Massachusetts General Hospital to help us with English writing.

REVIEWERS' COMMENTS:

Reviewer #1 (Remarks to the Author):

In the manuscript NO: Driving force of Bio-inspired Zero-waste Nanomotor and Its Endothelialization and Anticancer Effect, the authors claim a method for preparation of self-assembled NO-driven nanomotor, which can be tracked via fluorescence and has the ability to promote angiogenesis.

The topic is of general interest in the community, as it represents a new method to use the driving force of the system for biological applications. After the previous comments made by this reviewer, the manuscript was revised by the authors in a very complete way. I want to thank the authors for the effort made by answering all the points made in my previous document, and for completing the arguments in the manuscript in such way that their claims are better argued now. The language revision was also made, which has made the reading of the manuscript much easier this second time.

However, the novelty of the paper based on the topic chosen as main topic for this paper (nanomotors) is not the required for the high standards of nature communications, as I do not feel that the paper will influence thinking in this field.

As a last comment, I think there is still some statistical descriptions missing from the paper (many of the numbers mention in the text, have non standard deviations or errors mention to them), as well as I would suggest to shorten the manuscript, as is very long as it it.

Reviewer #2 (Remarks to the Author):

Authors developed here for the first time a new type of zero-waste and self-destroyed HLA nanomotors utilizing L-arginine as fuel, NO as driving force and L-citrulline as a beneficial by-product.

The revision made by the authors, have addressed carefully all my major concerns and improve significantly the quality and relevance of this work. Therefore I feel confident now to recommend its publication in Nature Communications in the present form.

REVIEWERS' COMMENTS:

Reviewer #1 (Remarks to the Author):

In the manuscript NO: Driving force of Bio-inspired Zero-waste Nanomotor and Its Endothelialization and Anticancer Effect, the authors claim a method for preparation of self-assembled NO-driven nanomotor, which can be tracked via fluorescence and has the ability to promote angiogenesis.

The topic is of general interest in the community, as it represents a new method to use the driving force of the system for biological applications. After the previous comments made by this reviewer, the manuscript was revised by the authors in a very complete way. I want to thank the authors for the effort made by answering all the points made in my previous document, and for completing the arguments in the manuscript in such way that their claims are better argued now. The language revision was also made, which has made the reading of the manuscript much easier this second time. However, the novelty of the paper based on the topic chosen as main topic for this paper (nanomotors) is not the required for the high standards of nature communications, as I do not feel that the paper will influence thinking in this field.

Response: Thanks so much for the reviewer's comments on our revised manuscript. We are pleased and grateful that the reviewer thinks that the manuscript was revised by us in a very complete way. We are very grateful to the reviewers for their efforts to improve the quality of the article.

As a last comment, I think there is still some statistical descriptions missing from the paper (many of the numbers mention in the text, have non standard deviations or errors mention to them), as well as I would suggest to shorten the manuscript, as is very long as it it.

Response: Thanks so much for the reviewer's kind reminding. We have added error bars for the numbers mentioned in the text as follows:

Figure 3. Movement behavior of the nanomotors. Time-lapse images (Supplementary Movie 2) displaying the tracking trajectories of (a) HLA₅, (b) HLA₁₀, (c) HLA₁₅ and (d) HLA₂₀ nanomotor in 10 s (20% H₂O₂, scale bar: 5 μm); Time-lapse images (Supplementary Movie 3) displaying the tracking trajectories of HLA₁₀ nanomotor in 10 s under (e) 5%, (f) 10%, and (g) 20% of H₂O₂ (Scale bar: 5 μm); (h) The consumption of L-arginine and (i) production of NO during motion process for HLA₁₀ nanomotor (20% H₂O₂); The speeds of (j) HLA_n nanomotor in 20% H₂O₂ and (k) HLA₁₀ nanomotor in different H₂O₂ concentrations.

Experimental data are mean +/- s.d. of samples in a representative experiment (n=3).

Meantime, the manuscript has been shortened as well. Thanks again for the reviewer's kind suggestions.

Reviewer #2 (Remarks to the Author):

Authors developed here for the first time a new type of zero-waste and self-destroyed HLA nanomotors utilizing L-arginine as fuel, NO as driving force and L-citrulline as a beneficial by-product.

The revision made by the authors, have addressed carefully all my major concerns and improve significantly the quality and relevance of this work. Therefore I feel confident now to recommend its publication in Nature Communications in the present form.

Response: Thanks so much for the reviewer's comments on our revised manuscript. We are very grateful to the reviewers for their efforts to improve the quality of the article.